# DunedinPACE, a DNA methylation biomarker of the pace of aging

Daniel W Belsky[1]\*, Avshalom Caspi[2], David L Corcoran[2], Karen Sugden[3], Richie Poulton[4], Louise Arseneault[5], Andrea Baccarelli[6], Kartik Chamarti[3], Xu Gao[7], Eilis Hannon[8], Hona Lee Harrington[3], Renate Houts[3], Meeraj Kothari[9], Dayoon Kwon[10], Jonathan Mill[8], Joel Schwartz[11], Pantel Vokonas[12], Cuicui Wang[11], Benjamin S Williams[3], Terrie E Moffitt[3]

[1]Department of Epidemiology & Butler Columbia Aging Center, Columbia University, New York, United States; [2]Center for Genomic and Computational Biology, Duke University, Durham, United States; [3]Department of Psychology and Neuroscience, Duke University, Durham, United States; [4]Department of Psychology, University of Otago, Otago, New Zealand; [5]Social, Genetic, and Developmental Psychiatry Centre, King's College London, London, United Kingdom; [6]Department of Environmental Health Sciences, Columbia University, New York, United States; [7]Department of Occupational and Environmental Health, Peking University, Beijing, China; [8]Complex Disease Epigenetics Group, University of Exeter, Exeter, United Kingdom; [9]Robert N Butler Columbia Aging Center, Columbia University, Brooklyn, United States; [10]Robert N Butler Columbia Aging Center, Columbia University, New York, United States; [11]Department of Environmental Health Sciences, Harvard TH Chan School of Public Health, Harvard University, Boston, United States; [12]Department of Medicine, VA Boston Healthcare System, Boston, United States

\*For correspondence: daniel.belsky@columbia.edu

## Abstract

**Background:** Measures to quantify changes in the pace of biological aging in response to intervention are needed to evaluate geroprotective interventions for humans. Previously, we showed that quantification of the pace of biological aging from a DNA-methylation blood test was possible (Belsky et al., 2020). Here, we report a next-generation DNA-methylation biomarker of Pace of Aging, DunedinPACE (for Pace of Aging Calculated from the Epigenome).

**Methods:** We used data from the Dunedin Study 1972–1973 birth cohort tracking within-individual decline in 19 indicators of organ-system integrity across four time points spanning two decades to model Pace of Aging. We distilled this two-decade Pace of Aging into a single-time-point DNA-methylation blood-test using elastic-net regression and a DNA-methylation dataset restricted to exclude probes with low test-retest reliability. We evaluated the resulting measure, named Dunedin-PACE, in five additional datasets.

**Results:** DunedinPACE showed high test-retest reliability, was associated with morbidity, disability, and mortality, and indicated faster aging in young adults with childhood adversity. DunedinPACE effect-sizes were similar to GrimAge Clock effect-sizes. In analysis of incident morbidity, disability, and mortality, DunedinPACE and added incremental prediction beyond GrimAge.

**Conclusions:** DunedinPACE is a novel blood biomarker of the pace of aging for gerontology and geroscience.

**Funding:** This research was supported by US-National Institute on Aging grants AG032282, AG061378, AG066887, and UK Medical Research Council grant MR/P005918/1.

## Editor's evaluation

This paper presents a new DNA methylation-based biomarker of aging: DunedinPACE. This biomarker is an updated version of DunedinPOAM, which was designed by the same group of authors to track an individual's Pace of Aging. It takes into account an additional measurement occasion (collected 20 years after inclusion) and only includes the most reliable DNA methylation probes, i.e. probes with little variation between technical replicates. DunedinPACE shows improved performance when compared to DunedinPOAM and can be used to complement previously generated DNA methylation-based biomarkers, such as GrimAge.

## Introduction

Guided by research on the cellular hallmarks of aging, model-organism studies are yielding treatments that slow the accumulation of deterioration in cells and organs (*Barzilai et al., 2018*; *Bellantuono, 2018*; *Campisi et al., 2019*). However, human geroprotector trials face a barrier when it comes to outcome measures: humans' long lifespan makes it prohibitively time-consuming to test whether a treatment extends healthspan (*Justice et al., 2018*). What is needed is a way to measure each clinical trial participant's personal pace of biological aging, before, during, after, and at long-term follow-up, to test whether a gero-protective therapy slows that pace, and whether benefits fade out (*Belsky et al., 2017*; *Moffitt et al., 2017*; *Sierra et al., 2021*).

An ideal research design for isolating biological patterns that differentiate faster and slower aging is longitudinal follow-up of a cohort of individuals who are all the same chronological age. We originally applied this design to test the geroscience-derived hypothesis that the process of biological aging was ongoing in healthy young adults and, critically, that it was already variable across individuals decades before most chronic diseases onset (*Belsky et al., 2015*; *Elliott et al., 2021b*). We studied a single-year birth cohort of all 1037 babies born in the city of Dunedin, New Zealand, during 1972–1973 and followed-up through midlife. Because aging can be understood as gradual, progressive deterioration simultaneously affecting different organ systems, we tracked declines in the cardiovascular, metabolic, renal, hepatic, immune, periodontal, and pulmonary systems of each participant using multiple biomarker measurements of each organ system, first at ages 26, 32, and 38 years, and subsequently including a fourth assessment at age 45 years. Modeling this one-of-a-kind dataset yielded a metric which quantified how slowly or rapidly each participant had been aging. We called this metric the Pace of Aging.

People with faster Pace of Aging tend to experience more rapid aging-related decline in physical and cognitive functions and develop early signs of brain aging (*Belsky et al., 2015*; *Elliott et al., 2021a*; *Elliott et al., 2021b*; *Rasmussen et al., 2019*). This evidence suggests that intervention to slow Pace of Aging could preserve functions lost with aging and extend healthspan. However, Pace of Aging takes many years to measure and requires data from several types of clinical and biological assays, limiting its utility in clinical trials. To address this limitation, we applied machine-learning tools to genome-wide DNA methylation data from Dunedin cohort members' blood samples to distill the multi-timepoint, multi-assay Pace of Aging measure into a single-timepoint, single-assay blood biomarker previously described in this journal: DunedinPoAm (for Dunedin Study Pace of Aging from methylation; *Belsky et al., 2020*). DunedinPoAm was designed to measure Pace of Aging biological change over time from a single blood sample. Like the original Pace of Aging, people with faster DunedinPoAm scores more often experienced declines in cognitive and physical functioning by midlife and showed more rapid facial aging *Belsky et al., 2020*; in older adults, faster DunedinPoAm predicted increased risk of disease and death (*Belsky et al., 2020*; *Graf et al., 2021*); in young people, experiences of early-life adversity were linked to faster DunedinPoAm (*Belsky et al., 2020*; *Raffington et al., 2021*).

DunedinPoAm provided proof of concept that Pace of Aging could be measured from a single blood test. But DunedinPoAm was based on biological change observed over just twelve years spanning the transition from young adulthood to midlife, limiting the scope of biological change that could be observed. And this biological change was measured from just three assessment waves, limiting precision. In addition, similar to other DNA methylation composites measured from microarray data, the measure had only moderate test-retest reliability (*Higgins-Chen et al., 2021*), limiting its value to clinical trials seeking to test within-participant changes from pre-treatment baseline to post-treatment

follow-up. To address these limitations, we incorporated new data from the Dunedin cohort extending Pace of Aging follow-up to include a fourth measurement occasion in the fifth decade of life (*Elliott et al., 2021b*) and restricted DNA-methylation data to exclude probes with poor reliability (*Sugden et al., 2020*) to refine our DNA-methylation biomarker of Pace of Aging. The longer follow-up period, which allowed us to observe more aging-related change in system integrity, and the additional measurement occasion included in analysis improve the power and precision of our modeling of longitudinal change in system integrity (*Brandmaier et al., 2018*; *Rast and Hofer, 2014*). The refined DNA-methylation dataset used for machine-learning improves reliability of measurement. We name the novel algorithm DunedinPACE, for Dunedin (P)ace of (A)ging (C)alculated from the (E)pigenome.

As a new DNA methylation measure of biological aging, DunedinPACE joins a well-established battery of DNA methylation measures known as 'clocks', so-named for their accurate prediction of chronological age (*Horvath and Raj, 2018*). However, DunedinPACE is distinct from the DNA methylation clocks in the method of its design and in its interpretation. DunedinPACE is derived from analysis of longitudinal data collected from a cohort of individuals who are all the same chronological age. It reflects differences between those individuals in the rate of deterioration in system integrity occurring over a fixed time interval, age 26 to age 45 years. It therefore has three distinguishing features that isolate aging from other confounds: (1) analysis of a single-year birth cohort distinguishes aging from confounding effects of exposures to factors that alter the methylome and that differ across generations, referred to hereafter as 'cohort effects'; (2) initiation of follow-up in young adults distinguishes aging from the effects of disease and excludes contamination from survival bias; and (3) focus on changes in multi-organ-system integrity during adult life distinguishes ongoing aging processes from deficits established early in development. The clocks, in contrast, were developed using a single cross-section of data from samples of mixed-age and older adults. They do not distinguish aging from cohort effects, are vulnerable to survival bias and contamination of aging signal by disease processes, and do not distinguish aging-related changes from prenatal/childhood deficits. DunedinPACE, therefore, represents a complementary tool with useful advantages for DNA methylation quantification of biological aging.

## Results

We developed the DunedinPACE measure reported here by analyzing the pace of biological aging in a 1972–1973 birth cohort (N = 1037), the Dunedin Study. This analysis consisted of two parts. The first part of analysis followed our original method (*Belsky et al., 2015*) to quantify Pace of Aging from two decades of longitudinal organ-system integrity data (*Elliott et al., 2021b*). First, we measured longitudinal changes in 19 biomarkers assessing cardiovascular, metabolic, renal, hepatic, immune, dental, and pulmonary systems, measured at ages 26, 32, 38 and, most recently in 2019, 45 years (*Supplementary file 1A* provides measurement details about the tracked biomarkers). Second, linear mixed-effects modeling was used to fit growth models to estimate each study member's personal rate of change for each of the 19 biomarkers. Third, these 19 personal rates of change were combined across biomarkers to calculate each Study member's personal Pace of Aging. In line with the geroscience hypothesis (*Kaeberlein, 2013*; *Kennedy et al., 2014*), which states that aging represents progressive decline across organ systems, we calculated each Study member's Pace of Aging as the sum of age-dependent annual changes across all biomarkers: Pace of Aging$_i$ = $\sum_{B=1}^{19} \mu_{1iB}$. The resulting Pace of Aging was then scaled to a mean of 1, so that it could be interpreted with reference to an average rate of 1 year of biological aging per year of chronological aging. Study members showed wide variation in their Pace of Aging (Mean = 1 biological year per chronological year, SD = 0.29). Over the two decades that we measured biological aging, the Study member with the slowest Pace of Aging aged by just 0.40 biological years per chronological year, while the Study member with the fastest Pace of Aging accrued 2.44 biological years per chronological year (*Elliott et al., 2021b*).

The second part of the analysis followed our original method to distill Pace of Aging into a blood test (*Belsky et al., 2020*). Using Illumina EPIC array DNA methylation data from blood collected at age 45, we developed DunedinPACE to provide a surrogate for the 20-year Pace of Aging. Briefly, we used elastic-net-regression (*Zou and Hastie, 2005*) to develop a DNA methylation algorithm to predict the 20-year Pace of Aging. Analysis included the subset of probes included on both the Illumina 450 k and EPIC arrays which we previously determined to have acceptable test-retest reliability (ICC >0.4, n = 81,2392) (*Sugden et al., 2020*). Following the method established by *Horvath, 2013*

and used in our prior analysis, we fixed the alpha parameter of the elastic net at 0.5. The final lamda value selected by the analysis was 0.042. The resulting algorithm included 173 CpG sites. The CpGs included in DunedinPACE are listed in *Supplementary file 1B*.

## DunedinPACE distills the 20-year pace of aging into a single-timepoint DNA methylation blood test

DunedinPACE demonstrated a high in-sample correlation with Pace of Aging to age 45 ($r = 0.78$, *Figure 1a*). The value indicates improved fit of the DunedinPACE model to the 20-year Pace of Aging as compared to the original DunedinPoAm version's fit to the 12-year Pace of Aging (in-sample correlation $r = 0.6$). Correlations among the 12- and 20-year Pace of Aging measures and the DunedinPoAm and DunedinPACE DNA methylation measures are shown in *Figure 1—figure supplement 1*.

Consistent with the high correlation of DunedinPACE with 20-year Pace of Aging, effect-sizes for associations of DunedinPACE with measures of physical and cognitive functioning and subjective signs of aging measured at age 45 in the Dunedin Study were similar to effect-sizes for the 20-year Pace of Aging. Effect-sizes for DunedinPACE are graphed in *Figure 1b*. Comparison with effect-sizes for the 20-year Pace of Aging reported previously (*Elliott et al., 2021b*; *Rasmussen et al., 2019*) are shown in *Figure 1—figure supplement 2*.

We also repeated analysis of decline in physical and cognitive functions included in our original 2020 *eLife* report of DunedinPoAm. This analysis tested associations between DunedinPACE and worsening in physical functions and subjective signs of aging between the age-38 and age-45 assessments and change in cognitive functioning from adolescent baseline to the age-45 assessment. Effect-sizes are graphed in *Figure 1c*.

Comparison of DunedinPACE effect-sizes for associations with function tests and subjective signs of aging to effect-sizes for the original DunedinPoAm and the DNA methylation clocks proposed by Horvath, Hannum et al., Levine et al. (PhenoAge Clock), and Lu et al. (GrimAge Clock) are shown in *Figure 1—figure supplements 3 and 4*. DunedinPACE effect-sizes are similar to or larger than those for DunedinPoAm and the GrimAge clock and larger than those for the other DNA methylation clocks. DunedinPACE and DunedinPoAm were developed from analysis of Pace of Aging in the Dunedin cohort; effect-sizes for these measures may be over-estimated in analyses of Dunedin data. We therefore conducted further validation analysis in four additional cohorts.

## DunedinPACE shows exceptional test-retest reliability

A Pace of Aging measure should have high test-retest reliability in order to be able to quantify change from pre-treatment baseline to post-treatment follow-up in intervention studies or exposure-related changes in longitudinal observational studies. We first evaluated test-retest reliability of DunedinPACE using a publicly available database of 36 replicate adult-blood-sample Illumina 450 k datasets (*Lehne et al., 2015*) (GEO Accession GSE55763). We estimated intraclass correlation coefficients (ICCs) for these replicates using mixed-effects regression. For the original DunedinPoAm, reliability was good (ICC = 0.89 95% CI [0.79–0.94]). For DunedinPACE, reliability was excellent (ICC = 0.96 [0.93–0.98]; *Figure 2*).We repeated this analysis in two additional databases of replicates reported by *Sugden et al., 2020*, a database of n = 28 replicate Illumina EPIC-array datasets and a database of n = 350 replicates in which one dataset was generated from the Illumina 450 k array and the other dataset was generated from the Illumina EPIC array. DunedinPACE ICCs were 0.97 [0.94–0.98] in the EPIC-EPIC database and 0.87 [0.82–0.90] in the 450k-EPIC database.

Comparison of DunedinPACE reliability to reliabilities for the original DunedinPoAm and the DNA methylation clocks are shown in *Figure 2—figure supplement 1*. DunedinPACE is as reliable as the GrimAge clock and more reliable in comparison to the other clocks, although methods have been proposed that may improve reliabilities for the other clocks (*Higgins-Chen et al., 2021*).

We conducted analysis to evaluate the extent to which improved fit to longitudinal Pace of Aging and better test-retest reliability for DunedinPACE as compared to the original DunedinPoAm might reflect (a) the four-time-point follow-up design of 20-year Pace of Aging and (b) the restriction of elastic net regression to develop DunedinPACE to probes that demonstrated higher test-retest reliabilities in blood in an independent sample. Results suggest that both four-time-points to measure Pace of Aging and restricting machine-learning analysis to CpG sites with higher test-retest reliability contributed to the technical improvement in DunedinPACE over DunedinPoAm (see **Appendix 1**).

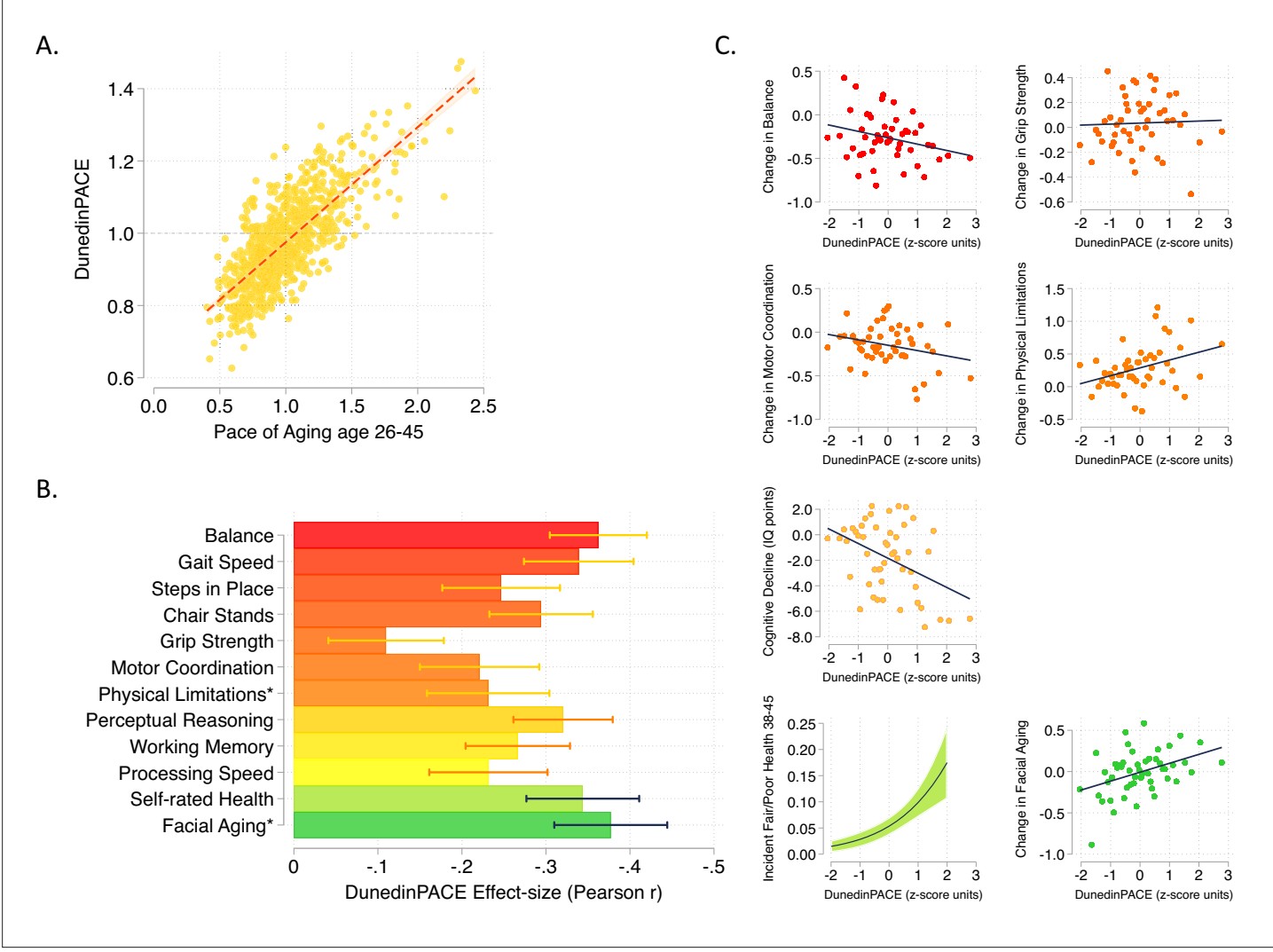

**Figure 1.** DunedinPACE correlation with 20-year Pace of Aging and association with decline in physical and cognitive functions and subjective signs of aging. Panel A shows the association of DunedinPACE with the 20-year Pace of Aging in the Dunedin Study cohort ($r = 0.78$). Panel B shows effect-sizes for associations of DunedinPACE with measures of physical and cognitive functioning and subjective signs of aging in Dunedin Study members at age 45 years. Colors indicate groupings of measures (physical functions in shades of orange, cognitive functions in shades of yellow, subjective signs of aging in shades of green). Stars next to labels for Physical Limitations and Facial Aging indicate that these meaures were reverse-coded for this analysis. Panel C shows binned scatterplots of associations of DunedinPACE with declines in physical and cognitive function and subjective signs of aging. Plotted points show average values for 'bins' of approximately 20 Study members. Regression slopes are estimated from the raw, un-binned data. In Panel C, y-axis scales are denominated in units of standard deviations computed from the baseline measurement for all outcomes except self-rated health, for which the y-axis shows probability of incident fair/poor self-rated health. Changes were calculated over the interval between age-38 and age-45 assessments for all measures except cognition, for which change was calculated over the interval between the age-13 and age-45 assessments. DunedinPACE and DunedinPoAm were developed from analysis of Pace of Aging in the Dunedin cohort; effect-sizes for these measures may be over-estimated.

The online version of this article includes the following figure supplement(s) for figure 1:

**Figure supplement 1.** Correlations among the 12-year and 20-year Pace of Aging measures and the DunedinPoAm and DunedinPACE DNA methylation measures.

**Figure supplement 2.** Effect-sizes for associations of DunedinPACE and the 20-year Pace of Aging with measures of physical and cognitive functioning and subjective signs of aging measured when Dunedin Study participants were aged 45 years.

**Figure supplement 3.** Effect-sizes for associations of DunedinPACE, DunedinPoAm, and the DNA methylation clocks proposed by *Horvath, 2013*, *Hannum et al., 2013*, *Levine et al., 2018* (PhenoAge), and *Lu et al., 2019* (GrimAge) with measures of physical and cognitive functioning and subjective signs of aging measured when Dunedin Study participants were aged 45 years.

*Figure 1 continued*

**Figure supplement 4.** Effect-sizes for associations of aging measures with measures of change in physical functioning and subjective signs of aging over ages 38–45 years and with cognitive functioning from adolescent baseline to age-45 follow-up.

## DunedinPACE is faster in chronologically and biologically older individuals

In demography, the pace of biological aging in a population can be estimated from the rate of increase in mortality risk from younger to older chronological ages (*Hägg et al., 2019*). In humans and many other species, the increase in mortality risk with advancing chronological age is nonlinear, suggesting that the pace of biological aging accelerates as we grow older (*Gompertz, 1997*; *Kirkwood, 2015*; *Olshansky and Carnes, 1997*). We tested if, consistent with this hypothesis, DunedinPACE was faster in chronologically older as compared to younger individuals in data from the Understanding Society Study (n = 1175,, age range 28–95). Chronologically older Understanding Society participants had faster DunedinPACE as compared to younger ones (r = 0.32, *Figure 3A*). This correlation was three-fold larger as compared to the correlation for the original DunedinPoAm (r = 0.11).

Between-individual correlation of older chronological age with faster DunedinPACE could be confounded by differences in exposure histories across birth cohorts, that is not an effect of within-individual aging (*Moffitt, 2020*). To rule out such confounding, we conducted analysis of within-individual change in DunedinPACE in the US Veterans Administration Normative Aging Study (NAS). Among NAS participants with repeated-measures DNA methylation data (n = 536), DunedinPACE increased by b = 0.021 (SE = 0.003) per 5 years of follow-up, or approximately 0.2 standard deviation units, similar to what we previously reported for the original DunedinPoAm (*Belsky et al., 2020*).

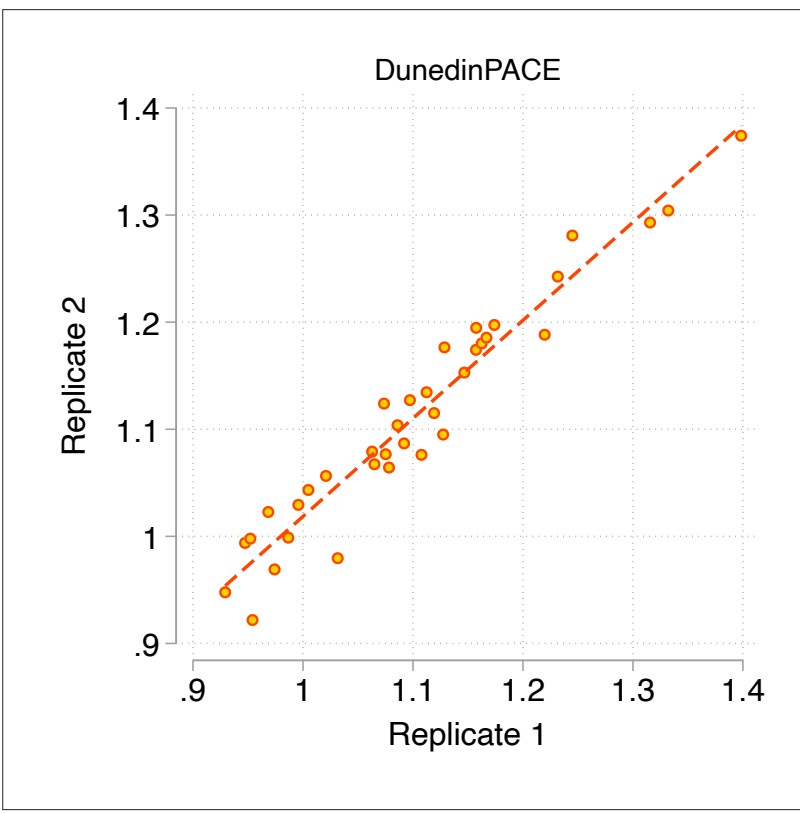

**Figure 2.** Test-retest reliability of DunedinPACE. The figure graphs DunedinPACE values for replicate Illumnina450k datasets for n = 36 individuals in the dataset published by Lehne and colleagues (*Lehne et al., 2015*) (GEO Accession GSE55763). The ICC for DunedinPACE in the Lehne dataset is 0.96 95% CI [0.92–0.98].

The online version of this article includes the following figure supplement(s) for figure 2:

**Figure supplement 1.** Reliability of DunedinPACE, DunedinPoAm, and DNA methylation clocks.

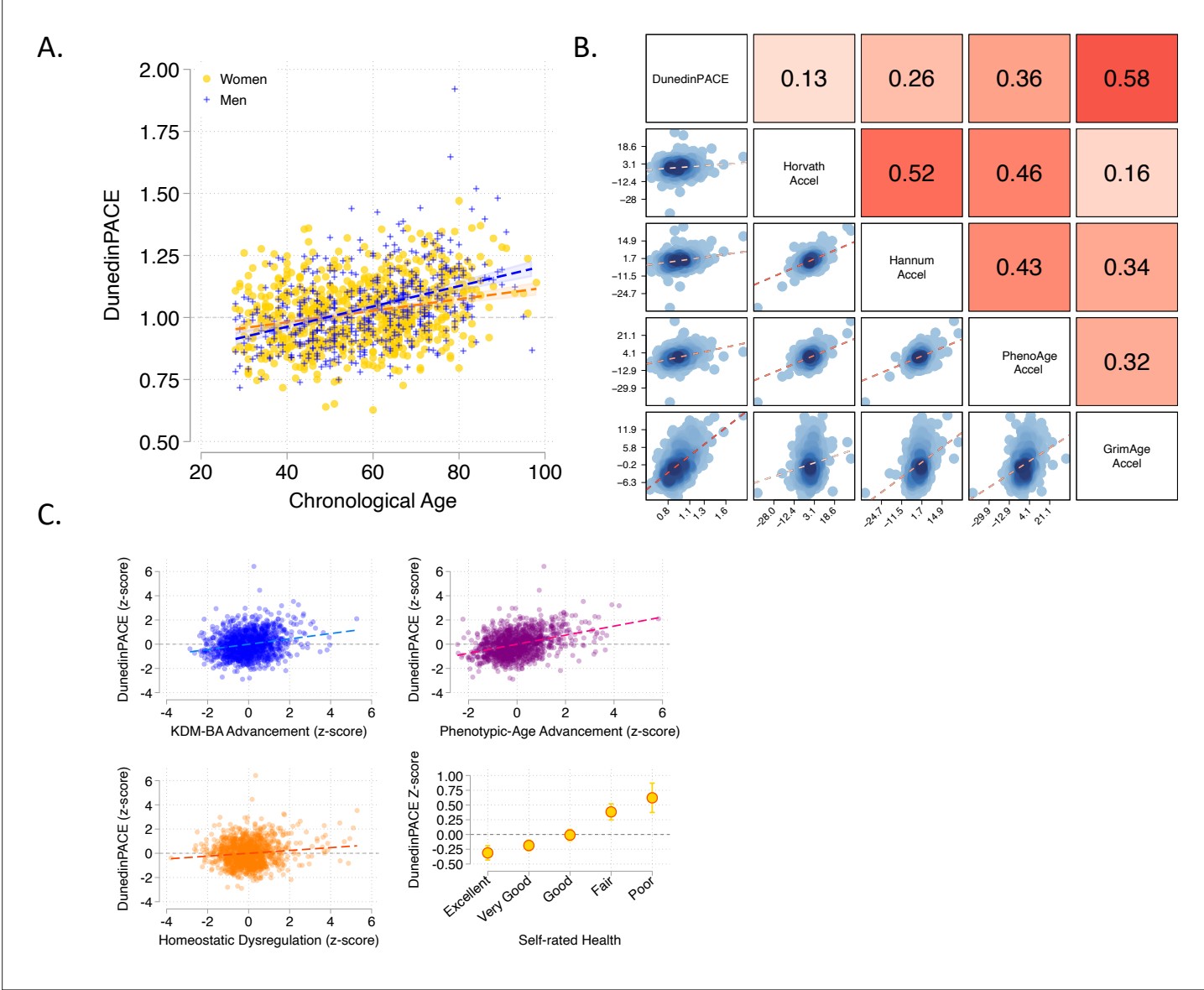

**Figure 3.** Associations of DunedinPACE with chronological age, epigenetic clocks, Physiology-based measures of biological age, and self-rated health in Understanding Society. Panel A shows a scatterplot and fitted slopes illustrating the association between chronological age (x-axis) and DunedinPACE (y-axis) in women and men in the Understanding Society sample. Data for women are plotted with yellow dots (orange slope) and for men with blue crosses (navy slope). The figure illustrates a positive association between chronological age and DunedinPACE (Pearson *r* = 0.32). Panel B shows a matrix of correlations and association plots among DunedinPACE and age-acceleration residuals of Horvath, Hannum, Levine-PhenoAge and Lu-GrimAge epigenetic clocks. The diagonal cells of the matrix list the DNA methylation measures. The half of the matrix below the diagonal shows scatter plots of associations. For each scatter-plot cell, the y-axis corresponds to the variable named along the matrix diagonal to the right of the plot and the x-axis corresponds to the variable named along the matrix diagonal above the plot. The half of the matrix above the diagonal lists Pearson correlations between the DNA methylation measures. For each correlation cell, the value reflects the correlation of the variables named along the matrix diagonal to the left of the cell and below the cell. Panel C graphs scatterplots of associations of DunedinPACE with three physiology-based measures of biological age (KDM Biological Age Advancement, *r* = 0.30 [0.24–0.36]; Phenotypic Age Advancement, *r* = 0.32 95% CI [0.26–0.38]; and Homeostatic Dysregulation *r* = 0.09 [0.03–0.16]) and a plot of DunedinPACE means and 95% confidence intervals by self-rated health category (*r* = 0.20 [0.15–0.26]).

The online version of this article includes the following figure supplement(s) for figure 3:

**Figure supplement 1.** Effect-sizes for associations of DunedinPACE, DunedinPoAm, and DNA methylation clocks with physiology-based measures of biological age and self-rated health.

DunedinPACE is faster in individuals measured to be biologically older using prior measures derived from DNA methylation and physiological data, and who report poorer self-rated health. We compared DunedinPACE with the epigenetic clocks proposed by *Hannum et al., 2013*; *Horvath, 2013*; *Levine et al., 2018*; *Lu et al., 2019*, with measurements of biological age derived from physiological data and with participants' subjective perceptions of their own health status in data from the UK-based Understanding Society Study (n = 1175,, mean age = 58, SD = 15, 42% male).

## Epigenetic clock analysis

For each epigenetic clock, we calculated the measure referred to as epigenetic 'age acceleration' by regressing participants' clock-estimated ages on their chronological ages and computing residual values. This measure is often used as an estimate of the aging rate. It quantifies the difference between the amount of aging expected based on a person's chronological age and the amount of aging observed based on DNA methylation. DunedinPACE was weakly correlated with age acceleration computed from the Horvath clock ($r = 0.13$) and somewhat more strongly correlated with age acceleration computed from the Hannum and Levine clocks (Hannum $r = 0.26$; PhenoAge $r = 0.36$). These correlations were similar to those reported on our 2020 *eLife* article for the original DunedinPoAm measure. For the GrimAge clock proposed by Lu et al., the correlation with the age-acceleration residuals was stronger ($r = 0.58$). Correlations are plotted in *Figure 3B*.

## Physiology-based measurements of biological age

We used published methods to measure participants' biological ages from blood-chemistry, blood-pressure, and lung function data collected at the same time as the DNA methylation data. We computed indices of biological age using three different methods: the Klemera and Doubal method (KDM) Biological Age estimates the age at which a person's physiology would appear normal in a reference population *Klemera and Doubal, 2006*; the Phenotypic Age method estimates the age at which a person's physiology-predicted mortality risk would be approximately normal in a reference population *Levine et al., 2018*; the homeostatic dysregulation method estimates the extent to which a person's physiology deviates from a state of health (*Cohen et al., 2013*). These algorithms generate correlated but distinct estimates conceptually related to biological age (*Hastings et al., 2019*; *Parker et al., 2020*). Algorithms were parameterized using a set of biomarkers available in Understanding Society and included in Levine's original implementation of the Klemera-Doubal Biological Age (*Levine, 2013*; albumin, alkaline phosphatase, blood urea nitrogen, creatinine, C-reactive protein, HbA1C, systolic blood pressure, forced expiratory volume in one second) and training data from the US National Health and Nutrition Examination Surveys (NHANES) (*Kwon and Belsky, 2021*). Consistent with analysis of epigenetic age acceleration, participants with more advanced biological aging measured from physiological data also showed faster DunedinPACE (KDM Biological Age Advancement $r = 0.30$; Phenotypic Age Advancement $r = 0.32$; Homeostatic Dysregulation $r = 0.09$; *Figure 3C*).

## Self-rated health

Participants who rated themselves as being in worse health also showed faster DunedinPACE. The difference in DunedinPACE between those rating their health as excellent and poor was large (Cohen's d = 0.74 [0.46–1.03]; *Figure 3C*).

Comparison of effect-sizes for associations with physiology-based measures of biological age and self-rated health between DunedinPACE and the original DunedinPoAm and the four DNA methylation clocks is shown in *Figure 3—figure supplement 1*. DunedinPACE effect-sizes are similar effect-sizes for DunedinPoAm and the GrimAge clock and larger in comparison to the other clocks.

## Faster DunedinPACE is associated with morbidity, disability, and mortality

We conducted analysis of morbidity and mortality in older men in the VA Normative Aging Study (NAS) and in the Framingham Heart Study Offspring cohort.

NAS analysis was conducted using the same methods reported in our original article (*Belsky et al., 2020*). Briefly, analysis included n = 771 older men with average age of 77 years (SD = 7) at DNA methylation measurement. Over follow-up from DNA collection in 1999–2013, 46% died and 23% were

diagnosed with a new chronic disease, including any of hypertension, type-2 diabetes, cardiovascular disease, chronic obstructive pulmonary disease, chronic kidney disease, and cancer. Participants with faster DunedinPACE were at increased risk for morbidity and mortality (for incident chronic disease morbidity, DunedinPACE HR = 1.23 [1.07–1.42]; for prevalent chronic disease morbidity, Dunedin-PACE RR = 1.16 [1.12–1.20]; for mortality, DunedinPACE HR = 1.26 95% CI [1.14–1.40], *Figure 4A*).

Framingham analysis included n = 2471 members of the Offspring cohort (54% women) with average age of 66 years (SD = 9) at DNA methylation measurement during 2005–2008. Over follow-up through 2018, **23%** died, **13%** were newly diagnosed with cardiovascular disease (CVD), and **6%** had a first stroke or transient ischemic attack (TIA). Disability follow-up was conducted through 2015 based on participant reports of limitations to activities of daily living (ADLs) on the Nagi, Katz, and Rosow-Brelsau scales. Participants with faster DunedinPACE at baseline were at increased risk for CVD, stroke/TIA, and mortality (CVD HR = 1.39 [1.26–1.54]; stroke/TIA HR = 1.37 [1.19–1.58]; mortality HR = 1.65 95% CI [1.51–1.79], *Figure 4B*). They were also more likely to develop disability (Nagi ADL IRR = 1.40 [1.19–1.65]; Katz ADL IRR = 1.33 [1.16–1.53]; Rosow-Breslau ADL IRR = 1.39 [1.24–1.56]).

Comparison of effect-sizes for associations with morbidity, disability, and mortality between DunedinPACE and the original DunedinPoAm and the DNA methylation clocks are shown in *Figure 4—figure supplements 1–4*. DunedinPACE effect-sizes were similar to those for DunedinPoAm and the GrimAge clock and larger than effect-sizes for the other measures.

## Childhood exposure to poverty and victimization is associated with faster DunedinPACE

A Pace of Aging measure should be sensitive to histories of exposure associated with earlier onset of morbidity and shorter lifespan and should provide indications of faster aging before clinical signs of aging-related disease develop, in time for prevention. To test if DunedinPACE indicated faster aging in young people with histories of exposure thought to shorten healthy lifespan, we analyzed data from n = 1,658 members of the E-Risk Longitudinal Study. The E-Risk Study follows a 1994–1995 birth cohort of same-sex twins. Blood DNA methylation data were collected when participants were aged 18 years. We analyzed two exposures associated with shorter healthy lifespan, childhood low socioeconomic status and childhood victimization. Socioeconomic status was measured from data on parents' education, occupation, and income (*Trzesniewski et al., 2006*). Victimization was measured from exposure dossiers compiled from interviews with the children's mothers and home-visit assessments conducted when the children were aged 5, 7, 10, and 12 (*Fisher et al., 2015*). The dossiers recorded children's exposure to domestic violence, peer bullying, physical and sexual harm by an adult, and neglect. 72% of the analysis sample had no victimization exposure, 21% had one type of victimization exposure, 4% had two types of exposure, and 2% had three or more types of exposure.

E-Risk participants with exposure to childhood poverty and victimization showed faster DunedinPACE (For low as compared to high childhood socioeconomic status, d = 0.38 95% CI [0.25–0.51]; for polyvictimization as compared to no victimization, d = 0.47 [0.17–0.77]; *Figure 5*).

Comparison of effect-sizes for associations with childhood poverty and victimization between DunedinPACE and the original DunedinPoAm and the DNA methylation clocks are shown in *Figure 5—figure supplement 1*. DunedinPACE effect-sizes were similar to those for DunedinPoAm and the GrimAge clock and larger than effect-sizes for the other clocks.

## Sensitivity analysis

Results for analysis of all studies including covariate adjustment for cell counts and smoking are reported in *Supplementary file 2*. Covariate adjustment for estimated cell counts and participant reports of smoking history resulted in modest attenuation of some effect-sizes, although DunedinPACE effect-sizes remained statistically different from zero in nearly all cases (associations with declines in balance and cognitive function were attenuated below the alpha = 0.05 threshold in Dunedin Study analysis including smoking as a covariate; differences between un-victimized and those with one type of victimization were attenuated below the alpha = 0.05 threshold in E-Risk Study analysis including smoking as a covariate; differences between high- and middle-SES groups were attenuated below the alpha = 0.05 threshold in E-Risk Study analysis excluding smokers). Compared to the original DunedinPoAm, covariate adjustment for smoking tended to have a smaller impact on effect-size estimates for DunedinPACE. In follow-up analysis of the Understanding Society Study, we found that

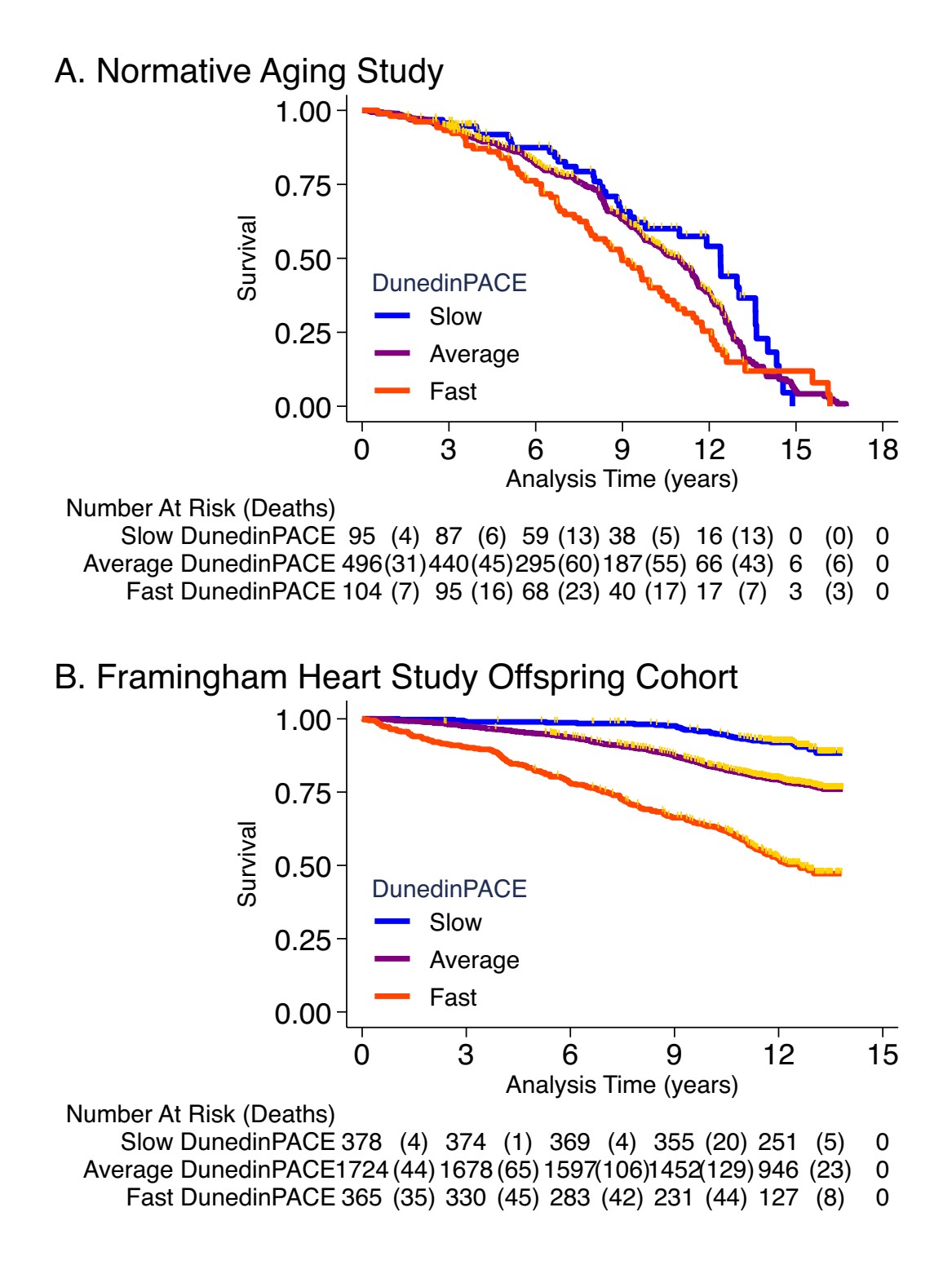

**Figure 4.** Association of DunedinPACE with mortality. Panel A shows mortality in the Normative Aging Study (NAS). Panel B shows mortality in the Framingham Heart Study Offspring Cohort. The figure plots Kaplan-Meier curves for three groups of participants in each of the two cohorts: those with DunedinPACE 1 SD or more below the mean ('slow' DunedinPACE, blue line); those with DunedinPACE within 1 SD of the mean ('average' DunedinPACE, purple line); and those with DunedinPACE 1 SD or more above the mean ('fast' DunedinPACE, red line). Censoring of participants prior

*Figure 4 continued on next page*

*Figure 4 continued*

to death is indicated with a gold hash marks. The table below the figure details the number of participants at risk per 3-year interval and, in parentheses, the number who died during the interval.

The online version of this article includes the following figure supplement(s) for figure 4:

**Figure supplement 1.** Effect-sizes for analysis of mortality in the Normative Aging Study and the Framingham Heart Study Offspring cohort.

**Figure supplement 2.** Effect-sizes for analysis of incident and prevalent chronic disease morbidity in the Normative Aging Study.

**Figure supplement 3.** Effect-sizes for analysis of time-to-diagnosis with cardiovascular disease (CVD) and time to stroke or transient ischemic attack (TIA) in the Framingham Heart Study Offspring Cohort.

**Figure supplement 4.** Effect-sizes for analysis of incident disability from repeated-measures of limitations to activities of daily living (ADLs).

participant-reported smoking history explained 30% of variance in DunedinPoAm, but only 13% of variance in DunedinPACE. Results were similar in analysis of the Framingham Heart Study Offspring Cohort ($R^2$ = 23% for DunedinPoAm vs. 12% for DunedinPACE).

## Incremental predictive value of DunedinPACE in an analysis of morbidity, disability, and mortality

DunedinPACE is distinct from DNA methylation clocks in the method of its development and in its interpretation. However, DunedinPACE is moderately correlated with some DNA methylation clocks, in particular GrimAge (see *Figure 3B*). We therefore conducted analysis to test if DunedinPACE contributed new information about health-span and lifespan over and above existing DNA methylation clocks. We tested if DunedinPACE associations with morbidity, disability, and mortality were statistically independent of each of the DNA methylation clocks within the Framingham Heart Study.

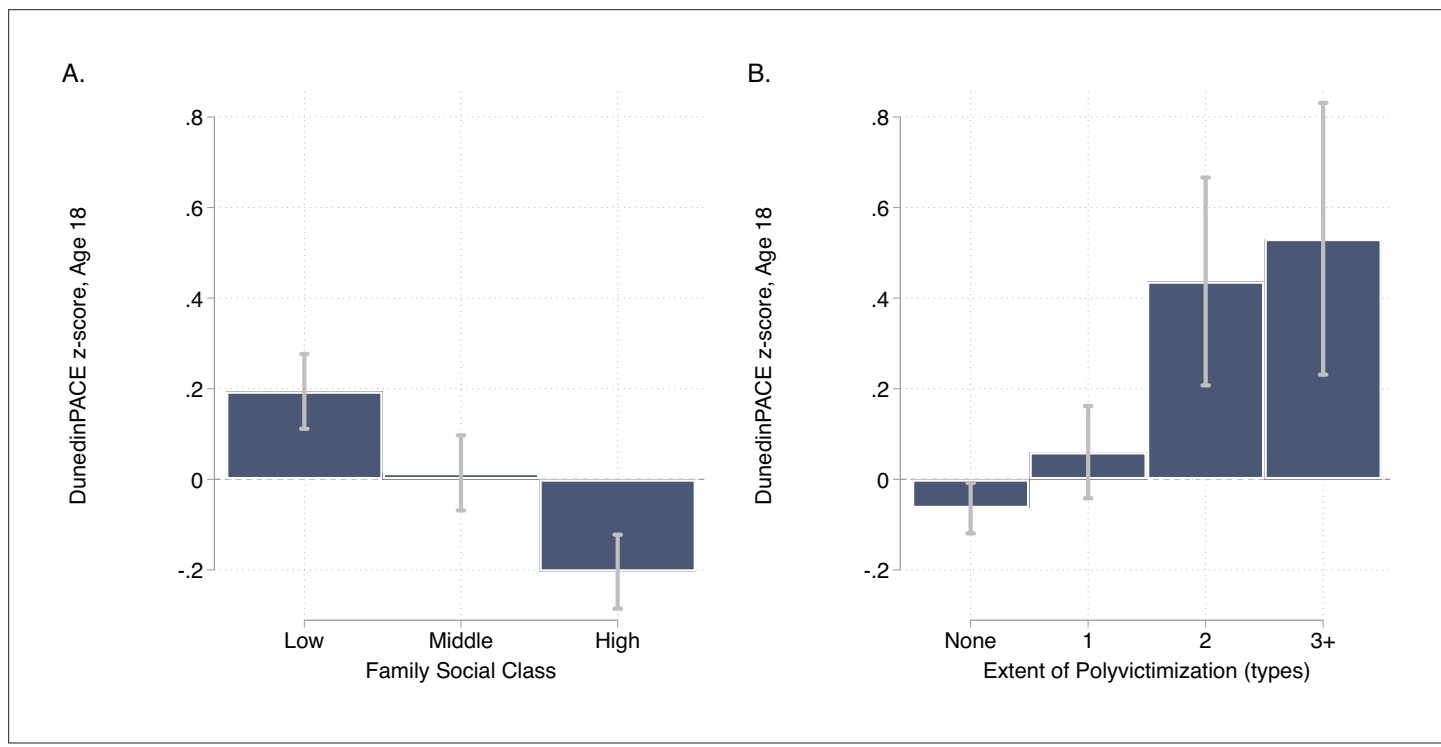

**Figure 5.** DunedinPACE levels by strata of childhood socioeconomic status (SES) and victimization in the E-Risk Study. Panel A (left side) plots means and 95% CIs for DunedinPACE measured at age 18 among E-Risk participants who grew up low, middle, and high socioeconomic status households. Panel B (right side) plots means and 95% CIs for DunedinPACE measured at age 18 among E-Risk participants who experienced 0, 1, 2, or 3 or more types of victimization through age 12 years.

The online version of this article includes the following figure supplement(s) for figure 5:

**Figure supplement 1.** Effect-sizes for associations of childhood socioeconomic conditions and childhood victimization history with DunedinPACE, DunedinPoAm, and DNA methylation clocks at age 18.

In the case of the GrimAge clock, which was developed to predict mortality using this Framingham dataset, this analysis provides an especially rigorous test.

DunedinPACE associations with health-span endpoints were little-changed by covariate adjustment for the Horvath, Hannum, or PhenoAge Clocks. In models adjusted for GrimAge, DunedinPACE associations with mortality, CVD, and disability were attenuated, but remained statistically different from zero (mortality HR = 1.24 [1.49–1.74], CVD HR = 1.18 [1.05–1.34], Nagi ADL IRR = 1.27 [1.02–1.58], Katz ADL IRR = 1.26 [1.02–1.54], Rosow-Breslau ADL IRR = 1.27 [1.08–1.50]); associations with stroke were similar to unadjusted models (HR = 1.33 [1.05–1.69]). Results for all models are reported in *Supplementary file 1C*. Thus, Dunedin PACE adds incremental prediction over and above all clocks studied here.

## Discussion

We developed a novel DNA methylation measure of the pace of biological aging from analysis of two decades of longitudinal data from members of the Dunedin Study 1972–1973 birth cohort. The new measure is named DunedinPACE. As in our original report (*Belsky et al., 2020*), we utilized a two-step approach that first modeled the Pace of Aging from within-individual changes in organ-system integrity markers and then distilled this longitudinal change measure into a single-time-point DNA-methylation blood test. DunedinPACE is therefore distinct from the DNA methylation clocks in both theory and method. The clocks estimate the progress of aging up to a cross-section in time. DunedinPACE is a DNA methylation estimate of the Pace of Aging, the ongoing rate of decline in system integrity.

The DunedinPACE measure advances DNA methylation measurement of the Pace of Aging for clinical trials of geroprotective therapies beyond our original DunedinPoAm measure in four ways: (1) Pace of Aging analysis included 20 years of follow-up (instead of 12); (2) Pace of Aging analysis included four time points of measurement (instead of three); (3) DNA methylation modeling of Pace of Aging excluded CpG sites for which probes were determined to have low test-retest reliability in blood datasets; (4) the algorithm to implement DunedinPACE includes a normalization step that allows DunedinPACE values for individual samples to be compared to the Dunedin Study cohort in which the measure was developed.

As a result of these advances, DunedinPACE provides a more precise measurement of the Pace of Aging as compared to the original DunedinPoAm and shows higher test-retest reliability. Interestingly, in contrast to reports for some epigenetic clocks (*Higgins-Chen et al., 2021*), the restriction of machine learning analysis to a more-reliably-measured subset of CpG sites on the Illumina EPIC array did not harm criterion validity of DunedinPACE; effect-sizes for prediction of morbidity and mortality were as large or larger in comparison to DunedinPoAm. Analysis of this more precise and reliable measurement revealed stronger associations with signs of aging. DunedinPACE showed a stronger correlation with chronological age as compared to the original DunedinPoAm, consistent with demographic evidence that biological processes of aging accelerate in later life (*Finch and Crimmins, 2016*). In analysis of construct and criterion validity, DunedinPACE was associated with functional decline in midlife adults; it was correlated with measures of biological age derived from blood chemistry and DNA methylation data, and with research participants' subjective perceptions of their own health; it was predictive of morbidity and mortality in midlife and older adults; and it indicated faster Pace of Aging in young adults with histories of exposure to poverty and victimization. Dunedin-PACE effect-sizes were similar to or larger than those for the original DunedinPoAm measure even as DunedinPACE was less influenced by participant smoking history. In analysis that took smoking history into account, DunedinPACE effect-sizes were more robust than those for DunedinPoAm, most notably in analysis of mortality in the Framingham Heart Study and of early-life adversity in the E-Risk cohort.

DunedinPACE effect-sizes in analysis of construct and criterion validity were also larger than those for benchmark DNA methylation clocks proposed by Horvath, Hannum et al., and Levine et al. (PhenoAge), and similar to those for the clock proposed by Lu et al. (GrimAge), and were robust to covariate adjustment for smoking and DNA methylation estimates of cell counts. Moreover, in Framingham Heart Study Offspring Cohort analysis of incremental predictive validity, Dunedin-PACE associations with morbidity, disability, and mortality were robust to covariate adjustment for the DNA methylation clocks, including GrimAge, although some associations were attenuated in

GrimAge-adjusted models. Framingham analysis represents a decisive test of value added by DunedinPACE over and above GrimAge because GrimAge was developed within the Framingham cohort.

In sum, DunedinPACE represents a novel measure of aging that is conceptually and empirically distinct from the DNA methylation clocks and that constitutes an advance on the original DunedinPoAm measure. DunedinPACE can complement existing DNA methylation measures of aging in analysis of observational studies and randomized trials to help advance the frontiers of geroscience.

We acknowledge limitations. Foremost, the Dunedin Study sample we analyzed to develop DunedinPACE is a relatively modestly sized cohort and is drawn from a single country. As additional cohorts develop the multi-time-point organ-system-function datasets to model Pace of Aging, integrating these data into DNA methylation algorithms to measure Pace of Aging may enhance precision. Nevertheless, DunedinPACE does show consistent evidence of criterion and construct validity in several additional datasets, boosting confidence in its external validity. In addition, our work on DunedinPACE thus far has not addressed population diversity in biological aging. The Dunedin cohort and the Understanding Society, NAS, Framingham, and E-Risk cohorts were mostly of white European descent. Follow-up of DunedinPACE in more diverse samples is needed to establish cross-population validity. Importantly, our original DunedinPoAm measure has been followed-up in diverse cohorts and has showed evidence of consistent validity across race/ethnic subgroups (*Crimmins et al., 2021*; *Graf et al., 2021*; *Raffington et al., 2021*; *Schmitz et al., 2021*). Criterion validity analyses conducted in this article do not consider specific diseases or causes of death because the cohorts used for follow-up are relatively small. Geroscience theory predicts aging should increase risk of chronic disease and death overall. Analysis of DunedinPACE in larger datasets with more deaths will allow for tests of whether particular incident diseases or causes of death drive associations. Comparative analyses in this article include only other DNA methylation measures of aging. Recent reports suggest valuable information about aging and healthspan may also be captured in metabolomic and proteomic datasets (*Deelen et al., 2019*; *Eiriksdottir et al., 2021*; *Jansen et al., 2021*; *Lehallier et al., 2019*; *Tanaka et al., 2020*). Future studies should compare DunedinPACE to measures derived from these biological levels of analysis. As with all proposed measures of biological aging, ultimately establishing DunedinPACE as a surrogate endpoint for healthspan will require evidence that it is modifiable by intervention and that intervention-induced changes result in changes in healthy-lifespan phenotypes (*Justice et al., 2018*; *Prentice, 1989*).

Within the bounds of these limitations, our analysis establishes DunedinPACE as a novel single-time-point measure that quantifies Pace of Aging from a blood test. It can be implemented in Illumina 450 k or in EPIC array data, making it immediately available for testing in a wide range of existing datasets as a complement to existing methylation measures of aging. Code to compute DunedinPACE is publicly available on GitHub (*Corcoran et al., 2022*). Through collaboration with ongoing cohort studies, DunedinPACE will soon be available from the US Health and Retirement Study and other studies. Because of its high test-retest reliability, DunedinPACE offers a unique measurement for intervention trials and natural experiment studies investigating how the rate of aging may be changed by behavioral or drug therapy, or by environmental modification.

# Materials and methods
## Data sources
Data were used from five studies: The Dunedin Study, the Understanding Society Study, the Normative Aging Study (NAS), the Framingham Heart Study, and the Environmental Risk (E-Risk) Longitudinal Twin Study. In addition, we accessed the Gene Expression Omnibus dataset GSE55763. The datasets and measures analyzed within each of them are described below.

### The Dunedin Study
The Dunedin Study is a longitudinal investigation of health and behavior in a complete birth cohort. Study members (N = 1037; 91% of eligible births; 52% male) were all individuals born between April 1972 and March 1973 in Dunedin, New Zealand (NZ), who were eligible based on residence in the province and who participated in the first assessment at age 3. The cohort represents the full range of socioeconomic status on NZ's South Island and matches the NZ National Health and Nutrition Survey on key health indicators (e.g. BMI, smoking, physical activity, GP visits) (*Poulton et al., 2015*).

The cohort is primarily white (93%) (**Poulton et al., 2015**). Assessments were carried out at birth and ages 3, 5, 7, 9, 11, 13, 15, 18, 21, 26, 32, 38 and, most recently, 45 years, when 94% of the 997 study members still alive took part. At each assessment, each study member is brought to the research unit for a full day of interviews and examinations. Study data may be accessed through agreement with the Study investigators (https://moffittcaspi.trinity.duke.edu/research-topics/dunedin). Dunedin Study measures of physical and cognitive functioning and subjective signs of aging are described in detail in **Supplementary file 1D**.

## GSE55763

GSE55763 is a publicly available dataset including technical replicate Illumina 450 k DNA methylation array data for 36 adult human samples (**Lehne et al., 2015**). Data are available from the Gene Expression Omnibus (https://www.ncbi.nlm.nih.gov/geo/query/acc.cgi?acc=GSE55763).

## Understanding Society

Understanding Society is an ongoing panel study of the United Kingdom population (https://www.understandingsociety.ac.uk/). During 2010–2012, participants were invited to take part in a nurse's exam involving a blood draw. Of the roughly 20,000 participants who provided clinical data in this exam, methylation data have been generated for just under 1200. We analyzed data from 1175 participants with available methylation and blood chemistry data. Documentation of the methylation (**Yanchun, 2017**) and blood chemistry (**University of Essex, 2013**) data resource is available online (here).

### Physiology-based biological age measures

We measured biological aging from blood chemistry, systolic blood pressure, and lung-function data using the algorithms proposed by Klemera and Doubal, Levine, and Cohen (**Cohen et al., 2013**; **Klemera and Doubal, 2006**; **Levine, 2013**; **Levine et al., 2018**), trained in data from US National Health and Nutrition Examination Surveys (NHANES; https://wwwn.cdc.gov/nchs/nhanes/Default.aspx) following the methods described by Hastings (**Hastings et al., 2019**) and using the software developed by Kwon (https://github.com/dayoonkwon/BioAge, copy archived at swh:1:rev:bac0d617f-4d04eb76350c0750cc367d8d1fd719e, **Kwon, 2022**). Following the procedure in our original eLife article (**Belsky et al., 2020**), we included 8 of 10 biomarkers included in Levine's original implementation of the Klemera-Doubal method Biological Age (**Levine, 2013**): albumin, alkaline phosphatase (log), blood urea nitrogen, creatinine (log), C-reactive protein (log), HbA1C, systolic blood pressure, and forced expiratory volume in 1 second ($FEV_1$). We omitted total cholesterol because of evidence this biomarker shows different directions of association with aging in younger and older adults (**Arbeev et al., 2016**). Cytomegalovirus optical density was not available in the Understanding Society database.

### Self-rated health

Understanding Society participants rated their health as excellent, very-good, good, fair, or poor. We standardized this measure to have Mean = 0, Standard Deviation = 1 for analysis.

### The Normative Aging Study

The Normative Aging Study (NAS) is an ongoing longitudinal study on aging established by the US Department of Veterans Affairs in 1963. Details of the study have been published previously (**Bell et al., 1972**). Briefly, the NAS is a closed cohort of 2280 male veterans from the Greater Boston area enrolled after an initial health screening to determine that they were free of known chronic medical conditions. Participants have been re-evaluated every 3–5 years on a continuous rolling basis using detailed on-site physical examinations and questionnaires. DNA from blood samples was collected from 771 participants beginning in 1999. We analyzed blood DNA methylation data from up to four repeated assessments conducted through 2013 (**Gao et al., 2019**; **Panni et al., 2016**). Of the 771 participants with DNA methylation data, n = 536 (70%) had data from two repeated assessments and n = 178 (23%) had data from three or four repeated assessments. We restricted the current analysis to participants with at least one DNA methylation data point. The NAS was approved by the Department

of Veterans Affairs Boston Healthcare System and written informed consent was obtained from each subject before participation.

## Mortality

Regular mailings to study participants have been used to acquire vital-status information and official death certificates were obtained from the appropriate state health department to be reviewed by a physician. Participant deaths are routinely updated by the research team and the last available update was on 31 December 2013. During follow-up, n = 354 (46%) of the 771 NAS participants died.

## Chronic disease morbidity

We measured chronic disease morbidity from participants medical histories and prior diagnoses (*Gao et al., 2021*; *Lepeule et al., 2018*). We counted the number of chronic diseases to compose an ordinal index with categories of 0, 1, 2, 3, or 4+ of the following comorbidities: hypertension, type-2 diabetes, cardiovascular disease, chronic obstructive pulmonary disease, chronic kidney disease, and cancer.

## The Framingham Heart Study

The Framingham Heart Study tracks the development of cardiovascular disease in three generations of families recruited in Framingham Massachusetts beginning in 1948 (*Tsao and Vasan, 2015*). We analyzed data from the Offspring Cohort, the second generation of study participants, who were recruited beginning in 1971. Blood samples used for DNA methylation analysis were collected from Offspring Cohort members at their 8th follow-up visit during 2005–2008. We analyzed data for n = 2471 cohort members with available DNA methylation and mortality data. This sample was 54% women and had an average age of 66 years (SD = 9) at the time blood draws for DNA methylation analysis were conducted. Data for the Framingham Study were obtained from dbGaP (phs000007.v32.p13).

### Mortality

Mortality ascertainment in our dataset extended through 2018 (dbGaP accession pht003317.v9.p13, dataset vr_survdth_2018_a_1268 s). Participants contributed a maximum of 14 years of follow-up time. A total of 575 deaths were recorded at an average age of 81 years (SD = 9).

### Cardiovascular disease (CVD)

CVD ascertainment in our dataset extended through 2018 (dbGaP accession pht003316.v9.p13, dataset vr_survcvd_2018_a_1267 s). N = 2,075 participants free of cardiovascular disease at baseline contributed a maximum of 14 years of follow-up time during which 327 incident cases of CVD were recorded at an average age of 77 years (SD = 6).

### Stroke/transient ischemic attack (TIA)

Stroke/TIA ascertainment in our dataset extended through 2018 (dbGaP accession pht006024.v4.p13, dataset vr_svstktia_2018_a_1270 s). N = 2364 participants with no history of stroke/TIA at baseline contributed a maximum of 14 years of follow-up time during which 149 events were recorded at an average age of 78 years (SD = 8).

### Disability

Disability was measured from participant reports about limitations to activities of daily living (ADLs) during interviews conducted at the 8th and 9th assessment waves. ADL information was collected using the Nagi, Katz, and Rosow-Breslau scales (*Katz et al., 1970*; *Nagi, 1976*; *Rosow and Breslau, 1966*). Scales are described in *Supplementary file 1E*. Incident disability was analyzed for participants free of disability at baseline (Nagi n = 1830; Katz n = 1826; Rosow-Breslau n = 1670). Over follow-up, n = 123 participants developed new limitations in Nagi ADLs; n = 181 developed new limitations in Katz ADLs; and n = 224 developed new limitations in Rosow-Breslau ADLs.

## The Environmental Risk Longitudinal Study

The Environmental Risk Longitudinal Twin Study (E-Risk) tracks the development of a birth cohort of 2232 British participants. The sample was drawn from a larger birth register of twins born in England and Wales in 1994–1995. Full details about the sample are reported elsewhere (*Moffitt and Team, 2002*). Briefly, the E-Risk sample was constructed in 1999–2000, when 1116 families (93% of

those eligible) with same-sex 5-year-old twins participated in home-visit assessments. This sample comprised 56% monozygotic (MZ) and 44% dizygotic (DZ) twin pairs; sex was evenly distributed within zygosity (49% male). Families were recruited to represent the UK population of families with newborns in the 1990s, on the basis of residential location throughout England and Wales and mother's age. Teenaged mothers with twins were over-selected to replace high-risk families who were selectively lost to the register through non-response. Older mothers having twins via assisted reproduction were under-selected to avoid an excess of well-educated older mothers. The study sample represents the full range of socioeconomic conditions in the UK, as reflected in the families' distribution on a neighborhood-level socioeconomic index (ACORN [A Classification of Residential Neighborhoods], developed by CACI Inc for commercial use): 25.6% of E-Risk families lived in 'wealthy achiever' neighborhoods compared to 25.3% nationwide; 5.3% vs 11.6% lived in 'urban prosperity' neighborhoods; 29.6% vs 26.9% lived in 'comfortably off' neighborhoods; 13.4% vs 13.9% lived in 'moderate means' neighborhoods, and 26.1% vs 20.7% lived in 'hard-pressed' neighborhoods. E-Risk underrepresents 'urban prosperity' neighborhoods because such households are likely to be childless.

Home-visits assessments took place when participants were aged 5, 7, 10, 12, and 18 years, when 93% of the participants took part. At ages 5, 7, 10, and 12 years, assessments were carried out with participants as well as their mothers (or primary caretakers); the home visit at age 18 included interviews only with participants. Each twin was assessed by a different interviewer. These data are supplemented by searches of official records and by questionnaires that are mailed, as developmentally appropriate, to teachers, and co-informants nominated by participants themselves. The Joint South London and Maudsley and the Institute of Psychiatry Research Ethics Committee approved each phase of the study. Parents gave informed consent and twins gave assent between 5 and 12 years and then informed consent at age 18. Study data may be accessed through the Study investigators (https://moffittcaspi.trinity.duke.edu/research-topics/erisk).

## Childhood Socioeconomic Status

Childhood Socioeconomic Status (SES) was defined through a standardized composite of parental income, education, and occupation (*Trzesniewski et al., 2006*). The three SES indicators were highly correlated ($r = 0.57$–$0.67$) and loaded significantly onto one factor. The population-wide distribution of the resulting factor was divided in tertiles for analyses.

## Childhood victimization

As previously described (*Danese et al., 2017*), we assessed exposure to six types of childhood victimization between birth to age 12: exposure to domestic violence between the mother and her partner, frequent bullying by peers, physical and sexual harm by an adult, and neglect.

## DNA methylation data

DNA methylation was measured from Illumina 450 k Arrays in GSE55763, NAS, Framingham, and E-Risk and from Illumina EPIC 850 k Arrays in the Dunedin Study and Understanding Society. DNA was derived from whole blood samples in all studies. Dunedin Study blood draws were conducted at the cohort's age-38 assessment during 2010–12 for the prior DunedinPoAm measure, and at the cohort's age-45 assessment during 2017–2019 for the new DunedinPACE measure. Understanding Society blood draws were conducted in 2012. NAS blood draws were conducted during 1999–2013. Framingham blood draws were conducted during 2005–2008. E-Risk blood draws were conducted at the cohort's age-18 assessment during 2012–13. Dunedin methylation assays were run by the Molecular Genomics Shared Resource at Duke Molecular Physiology Institute, Duke University (USA). Understanding Society and E-Risk assays were run by the Complex Disease Epigenetics Group at the University of Exeter Medical School (UK) (https://www.epigenomicslab.com). NAS methylation assays were run by the Genome Research Core of the University of Illinois at Chicago. Framingham methylation assays were run by the University of Minnesota and John's Hopkins University (dbGaP phs000724.v9.p13). Processing protocols for the methylation data from all studies have been described previously (*Dai et al., 2017*; *Hannon et al., 2018*; *Marzi et al., 2018*; *Panni et al., 2016*).

## DNA methylation clocks

We computed the methylation clocks proposed by *Horvath, 2013*, *Hannum et al., 2013*, *Levine et al., 2018*, and *Lu et al., 2019* using the methylation data provided by the individual studies and the Horvath Lab's webtool (http://dnamage.genetics.ucla.edu/new).

## DunedinPACE

The new Dunedin Pace of Aging methylation algorithm (DunedinPACE) was developed using elastic-net regression analysis carried out in the Dunedin Study, as described in detail in the Results. The criterion variable was 20-year Pace of Aging. Development of the 20-year Pace of Aging is described in detail elsewhere (*Elliott et al., 2021b*). Briefly, we conducted mixed-effects growth modeling of longitudinal change in 19 biomarkers measuring integrity of the cardiovascular, metabolic, renal, hepatic, pulmonary, periodontal, and immune systems. Biomarkers were measured at the age 26, 32, 38, and 45 assessments. Biomarkers are listed in *Supplementary file 1A*. For each biomarker, we estimated random slopes quantifying each participant's own rate of change in that biomarker. We then composited, the 19 slopes across the biomarkers to calculate a participant's Pace of Aging. Pace of Aging was scaled in units representing the mean trend in the cohort, that is the average physiological change occurring during one calendar year (N = 955, M = 1, SD = 0.29). Of the N = 818 Dunedin Study members with methylation data at age 45, N = 817 had measured Pace of Aging (M = 0.99, SD = 0.30). This group formed the analysis sample to develop DunedinPACE.

To compute DunedinPACE in the GSE55763, Understanding Society, NAS, Framingham, and E-Risk datasets, we applied the scoring algorithm derived from elastic net regression in the Dunedin Study. CpG weights for the scoring algorithm are provided in *Supplementary file 2*. To apply the scoring algorithm, a panel of 20,000 probes that represent the underlying distribution of all probes included in the analysis to develop DunedinPACE are drawn from the target dataset. This new dataset is then quantile-normalized to match the reference distribution from the Dunedin Study. The set of 20,000 probes is composed of the 173 probes that make up the DunedinPACE algorithm and 19,827 additional probes selected at random from the full set of probes included in analysis to develop DunedinPACE i.e probes passing quality control checks, which are included on Illumina 450 k and EPIC arrays, and which were reliable at an ICC threshold of 0.4 in the analysis by *Sugden et al., 2020*.

## Statistical analysis

We conducted analysis of Dunedin, GSE55763, Understanding Society, NAS, Framingham, and E-Risk, data using regression models. We analyzed continuous outcome data using linear regression. We analyzed count outcome data using Poisson and negative binomial regression. We analyzed time-to-event outcome data using Cox proportional hazard regression. To estimate intraclass correlation coefficients for technical replicates in GSE55763, we used mixed-effects regression models. For analysis of repeated-measures longitudinal DNA methylation data in the NAS, we used generalized estimating equations to account for non-independence of repeated observations of individuals (*Ballinger, 2016*), following the method in previous analysis of those data (*Gao et al., 2018*), and econometric fixed-effects regression (*Wooldridge, 2012*) to test within-person change over time. For analysis in E-Risk, which include data on twin siblings, we clustered standard errors at the family level to account for non-independence of data. For regression analysis, methylation measures were adjusted for batch effects by regressing the measure on batch controls and predicting residual values. Dunedin Study, Understanding Society, Framingham, and E-Risk analyses included covariate adjustment for sex (the NAS included only men). Understanding Society, NAS, and Framingham analyses included covariate adjustment for chronological age. (Dunedin and E-Risk are birth-cohort studies and participants are all the same chronological age.) Sensitivity analyses testing covariate adjustment for leukocyte distributions and smoking are reported in *Supplementary file 2*. Leukocyte distributions were measured from complete blood count data in the Dunedin Cohort and were estimated from DNA methylation data for other cohorts.

## Code for Analysis

Code is available at https://github.com/danbelsky/DunedinPACE-eLife-2022 (copy archived at swh:1:rev:e38233555b70c34f84f0b73e18f1a6bc4cb0852e, *Belsky, 2022a*).

Code to calculate DunedinPACE from Illumina 450 k or Epic Array Data. Code is available at https://github.com/danbelsky/DunedinPACE, (copy archived at swh:1:rev:a878c3706aec2ac41bc029704cef-321f7c0a4e38, *Belsky, 2022b*).

## Acknowledgements

This research was supported by US-National Institute on Aging grants AG032282, AG061378, AG066887, and UK Medical Research Council grant MR/P005918/1.

The Dunedin Multidisciplinary Health and Development Research Unit is supported by the New Zealand Health Research Council Programme Grant (16-604), and the New Zealand Ministry of Business, Innovation and Employment (MBIE). We thank the Dunedin Study members, Unit research staff, and Study founder Phil Silva.

Understanding Society data come from The UK Household Longitudinal Study, which is led by the Institute for Social and Economic Research at the University of Essex and funded by the Economic and Social Research Council (ES/M008592/1). Information on how to access the data can be found on the Understanding Society website https://www.understandingsociety.ac.uk. Data governance was provided by the METADAC data access committee, funded by ESRC, Wellcome, and MRC (2015–2018: MR/N01104X/1; 2018–2020: ES/S008349/1).

The Normative Aging Study is supported by the National Institute of Environmental Health Sciences (grants P30ES009089, R01ES021733, R01ES025225, and R01ES027747). The VA Normative Aging Study is supported by the Cooperative Studies Program/Epidemiology Research and Information Center of the U.S. Department of Veterans Affairs competeand is a component of the Massachusetts Veterans Epidemiology Research and Information Center, Boston, Massachusetts.

The Framingham Heart Study is conducted and supported by the National Heart, Lung, and Blood Institute (NHLBI) in collaboration with Boston University (Contract No. N01-HC-25195, HHSN268201500001I and 75N92019D00031). This manuscript was not prepared in collaboration with investigators of the Framingham Heart Study and does not necessarily reflect the opinions or views of the Framingham Heart Study, Boston University, or NHLBI.

The E-Risk Study is supported by the UK Medical Research Council (grant G1002190), the US National Institute of Child Health and Development (grant HD077482), and the Jacobs Foundation. The generation of DNA methylation data was supported by the American Asthma Foundation.

This work used a high-performance computing facility partially supported by grant 2016-IDG-1013 (HARDAC+: Reproducible HPC for Next-generation Genomics") from the North Carolina Biotechnology Center. We would like to acknowledge the assistance of the Duke Molecular Physiology Institute Molecular Genomics Core for the generation of data for the manuscript.

AC, DC, DWB, KS, RP, and TEM are listed as inventors on a Duke University and University of Otago invention that was licensed to a commercial entity.

## Additional information

### Competing interests

Daniel W Belsky, Avshalom Caspi, David L Corcoran, Karen Sugden, Richie Poulton, Terrie E Moffitt: is listed as an inventor on a Duke University and University of Otago invention that was licensed to a commercial entity. The other authors declare that no competing interests exist.

### Funding

| Funder | Grant reference number | Author |
|---|---|---|
| National Institute on Aging | AG032282 | Terrie E Moffitt |
| Medical Research Council | MR/P005918/1 | Terrie E Moffitt |
| National Institute on Aging | AG066887 | Daniel W Belsky |

| Funder | Grant reference number | Author |
|---|---|---|
| National Institute on Aging | AG061378 | Daniel W Belsky |
| New Zealand Health Research Council | 16-604 | Richie Poulton |
| New Zealand Ministry of Business, Innovation and Employment (MBIE) | | Richie Poulton |

The funders had no role in study design, data collection and interpretation, or the decision to submit the work for publication.

## Author contributions

Daniel W Belsky, Conceptualization, Data curation, Formal analysis, Funding acquisition, Investigation, Methodology, Project administration, Resources, Software, Supervision, Validation, Visualization, Writing – original draft, Writing – review and editing; Avshalom Caspi, Conceptualization, Funding acquisition, Investigation, Project administration, Supervision, Writing – review and editing; David L Corcoran, Formal analysis, Methodology, Software, Writing – review and editing; Karen Sugden, Data curation, Methodology, Writing – review and editing; Richie Poulton, Data curation, Funding acquisition, Project administration, Resources, Writing – review and editing; Louise Arseneault, Andrea Baccarelli, Eilis Hannon, Hona Lee Harrington, Jonathan Mill, Joel Schwartz, Pantel Vokonas, Benjamin S Williams, Data curation, Writing – review and editing; Kartik Chamarti, Cuicui Wang, Writing – review and editing; Xu Gao, Renate Houts, Dayoon Kwon, Formal analysis, Writing – review and editing; Meeraj Kothari, Formal analysis, Visualization, Writing – review and editing; Terrie E Moffitt, Conceptualization, Data curation, Funding acquisition, Project administration, Resources, Supervision, Writing – review and editing

## Author ORCIDs

Daniel W Belsky http://orcid.org/0000-0001-5463-2212
Eilis Hannon http://orcid.org/0000-0001-6840-072X
Jonathan Mill http://orcid.org/0000-0003-1115-3224

## Decision letter and Author response

Decision letter https://doi.org/10.7554/eLife.73420.sa1
Author response https://doi.org/10.7554/eLife.73420.sa2

---

# Additional files

## Supplementary files

• Supplementary file 1. Supplementary file 1A-E. (A) Measurements included in modeling Pace of Aging in the Dunedin Study Birth Cohort. We measured Pace of Aging from repeated assessments of a panel of 19 biomarkers: Body mass index (BMI), Waist-hip ratio, Glycated hemoglobin, Leptin, Blood pressure (mean arterial pressure), Cardiorespiratory fitness (VO$_2$Max), Forced vital capacity ratio (FEV$_1$/FVC), Forced expiratory volume in one second (FEV$_1$), Total cholesterol, Triglycerides, High density lipoprotein (HDL), Lipoprotein(a), Apolipoprotein B100/A1 ratio, estimated Glomerular Filtration Rate (eGFR), Blood Urea Nitrogen (BUN), High Sensitivity C-reactive Protein (hs-CRP), White blood cell count, mean periodontal attachment loss (AL), and the number of dental-caries-affected tooth surfaces (tooth decay). This list includes two new biomarkers, leptin and caries, not included in the original Pace of Aging (*Belsky et al., 2015*), both of which have now been assessed at multiple waves, allowing growth curve modeling. Telomere length was dropped because of an emerging and yet-unresolved field-wide debate about its measurement (*Nettle et al., 2021*). All other biomarkers were the same. The 20 year Pace of Aging measure is described in detail elsewhere (*Elliott et al., 2021b*). (B) CpGs included in the DunedinPACE DNA methylation algorithm, their mean values in the Dunedin Study, and the coefficients for their weights in the DunedinPACE algorithm. (C) Independent associations of DunedinPACE with mortality, morbidity, and disability in models including covariate adjustment for DNA methylation clocks. The table reports effect-sizes for DunedinPACE and DNA methylation clocks from time-to-event analysis of mortality, cardiovascular disease (CVD), and stroke or transient ischemic attack (TIA) and repeated-measures analysis of incident limitations to activities of daily living (ADLs) in the Framingham Heart

Study Offspring cohort. Each model includes DunedinPACE, one of the DNA methylation clocks, and covariates for age and sex. Time-to-event model effect-sizes are reported as hazard ratios (HR). Repeated-measures model effect-sizes are reported as incidence-rate ratios (IRR). (D) Physical and cognitive functioning and subjective signs of aging measures in the Dunedin Study. (E) Items included in Nagi, Katz, and Rosow-Breslau scales of limitations to Activities of Daily Living (ADLs).

• Supplementary file 2. Effect-sizes for analysis of DunedinPACE, DunedinPoAm, and DNA methylation clocks in models with covariate adjustment for age and sex only and for models with additional covariate adjustment for DNA methylation estimated cell counts and for smoking. Effect-sizes are reported for a one standard deviation increment of DunedinPACE. Outcome variables in Dunedin Study, Understanding Society Study, and E-Risk Study (Panels A, B, C, and E) were z-transformed for analysis; effect-sizes are interpretable as Pearson r or Cohen's d. For Normative Aging Study and Framingham Analysis (Panel C), effect-sizes for time-to event outcomes are reported as hazard ratios (HR), effect-sizes for prevalent chronic disease are reported as Risk Ratios (RR), and effect-sizes for incident disability are reported as incidence rate ratios (IRR). (A) Effect-sizes for associations of DunedinPACE, DunedinPoAm, and DNA methylation clocks with functional assessments at age 45 in the Dunedin Study. Effect-sizes are standardized coefficients from linear regressions of the outcomes on the aging measures. The models adjusted for cell counts (middle column of results) were adjusted for cell counts measured from complete blood count data (white blood cell count and percentages of lymphocytes, monocytes, neutrophils, eosinophils, and basophils). The models adjusted for smoking (left-most column of results) included a covariate for number of pack-years smoked. (B) Effect-sizes for associations of DunedinPACE, DunedinPoAm, and DNA methylation clocks with functional decline in the Dunedin Study. Decline was measured as change between the age-38 and age-45 assessments for all measures except cognition, for which change was computed over the interval beginning with an adolescent baseline test at age 13 and ending with the most recent cognitive assessment at age 45. Effect-sizes are coefficients from linear regressions of change scores on aging measures for all outcomes except incident fair/poor health, for which effect-sizes are risk ratios (RR) estimated from Poisson regression of incidence on the aging measures. Change scores were computed by standardizing baseline and follow-up measurements using the mean and standard deviation (SD) of the baseline assessment and computing the difference in standardized scores. A single unit of change therefore corresponds to 1 SD of variation at baseline. Effect-sizes are reported for a one SD difference in the aging measures. The models adjusted for smoking (left-most column of results) included a covariate for number of pack-years smoked. (C) Effect-sizes for associations of DunedinPACE, DunedinPoAm, and DNA methylation clocks with clinical-biomarker measures of biological age and self-rated health in the Understanding Society Study. Effect-sizes are standardized coefficients from linear regressions of the outcomes on the DNA methylation measures of aging. The models adjusted for smoking included covariates for smoking status and current smoking quantity (cigarettes per day). (D) Effect-sizes for associations of DunedinPACE, DunedinPoAm, and DNA methylation clocks with mortality, morbidity, and disability in the Normative Aging Study and the Framingham Heart Study Offspring Cohort. Effect-sizes are hazard ratios (HR) for time-to-event analyses and relative risks (RR) and incidence rate ratios (IRR) for repeated-measures analyses. Analysis of mortality was conducted in both cohorts. Analysis of incident and prevalent chronic disease morbidity was conducted only in the Normative Aging Study. Time-to-event analysis of cardiovascular disease (CVD) and stroke/transient ischemic attack (TIA) and repeated measures analysis of incident disability were conducted only in the Framingham Heart Study Offspring Cohort. The models adjusted for smoking included covariates for smoking history (pack-years smoked) in the Normative Aging Study and for smoking status and current smoking quantity (cigarettes per day) in the Framingham Heart Study. (E) Effect-sizes for associations of childhood socioeconomic status (SES) and childhood victimization with aging measures in the E-Risk Study. Effect-sizes are standardized coefficients (Cohen's d) from linear regressions of the aging measures on exposure groups (low and middle SES vs. high SES; one, two, and three or more types of victimization vs. no victimization). Models included covariate adjustment for sex. Standard errors were clustered at the family level to account for non-independence of sibling data.

• Transparent reporting form

### Data availability

Datasets are available from the data owners. Data from the Dunedin and E-Risk Study can be accessed through agreement with the Study investigators. Instructions are available at https://sites.google.com/site/moffittcaspiprojects/. The data access application form can be downloaded here: https://sites.google.com/site/moffittcaspiprojects/forms-for-new-projects/concept-paper-template. Data from the Understanding Society Study is available through METADAC at https://

www.metadac.ac.uk/ukhls/. All details are on the Metadac website (https://www.metadac.ac.uk/data-access-through-metadac/). The data access application form can be found here https://www.metadac.ac.uk/files/2019/02/v2.41-UKHLS-METADAC-application-form-2019-2hak8bv.docx. Data from the Normative Aging Study were obtained from the Study investigators. Data are accessible through dbGaP, accession phs000853.v1.p1. Data from the Framingham Heart Study were obtained from dbGaP, accession phs000007.v32.p13. GSE55763 is a publicly available dataset available from the Gene Expression Omnibus (https://www.ncbi.nlm.nih.gov/geo/query/acc.cgi?acc=GSE55763).

The following previously published datasets were used:

| Author(s) | Year | Dataset title | Dataset URL | Database and Identifier |
|---|---|---|---|---|
| Baccarelli AA, Schwartz J | 2015 | The Normative Aging Study | https://www.ncbi.nlm.nih.gov/projects/gap/cgi-bin/study.cgi?study_id=phs000853.v1.p1 | dbGaP, phs000853.v1.p1 |
| The National Heart Lung and Blood Institute, Boston University | 2020 | The Framingham Heart Study | https://www.ncbi.nlm.nih.gov/projects/gap/cgi-bin/study.cgi?study_id=phs000007.v32.p13 | NCBI dbGaP, phs000007.v32.p13 |
| Lehne | 2021 | A coherent approach for analysis of the Illumina HumanMethylation450 BeadChip improves data quality and performance in epigenome-wide association studies | https://www.ncbi.nlm.nih.gov/geo/query/acc.cgi?acc=GSE55763 | NCBI Gene Expression Omnibus, GSE55763 |

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

# Appendix 1

Correlation with criterion Pace of Aging and test-retest reliability for DunedinPACE, original DunedinPoAm, and alternative versions of these algorithms developed using different sets of CpG sites.

DunedinPACE and DunedinPoAm were both developed from machine-learning analysis fitting DNA methylation data to Pace of Aging measures derived from longitudinal analysis of organ-function and blood chemistry data within the Dunedin Study.

There are two key differences between the measures. First, the Pace of Aging measure modeled to develop DunedinPoAm included data from only three repeated measures spanning 12 years of follow-up. In contrast, the Pace of Aging measure modeled to develop DunedinPACE included data from four repeated measurements collected over 20 years of follow-up. Second, DunedinPoAm was developed from a DNA methylation dataset including all probes included on both the Illumina450k and EPIC arrays. In contrast, the DNA methylation dataset used to develop DunedinPACE was restricted to a subset of CpG sites meeting a test-retest reliability threshold.

These differences should have two consequences for DunedinPACE. First, the inclusion of a 4th timepoint in Pace of Aging analysis should increase precision of measurement and result in improved fit of DNA methylation data to the 4-timepoint Pace of Aging criterion as compared to the 3-timepoint version. (A less precise Pace of Aging will be harder for the elastic net machine learning analysis to model since more of the variation will be random noise.) Second, the restriction of machine learning to more reliable CpG sites should result in improved test-retest reliability.

To further test these improvements, we conducted a sensitivity analysis. We repeated the machine-learning analysis to construct DunedinPACE using the unrestricted set of CpG sites included in the original DunedinPoAm analysis. We refer to the resulting measure as "All-probes DunedinPACE". In parallel, we repeated the machine-learning analysis to construct DunedinPoAm using the subset of more-reliable CpG sites included in the original DunedinPACE analysis. We refer to the resulting measure as "Reliable Probes DunedinPoAm". We then repeated our analyses testing correlation of the DNA methylation measures with the criterion Pace of Aging and our analysis of test-retest reliability.

Consistent with the expectation that a 4th timepoint would increase precision of Pace of Aging measurement and result in improved criterion fit, both versions of DunedinPACE showed stronger correlation with 4-timepoint Pace of Aging than did the versions of DunedinPoAm with 3-timepoint Pace of Aging. For algorithms trained on all probes, $r = 0.87$ DunedinPACE vs. $r = 0.58$ for original DunedinPoAm. For algorithms trained on reliable probes, $r = 0.77$ for DunedinPACE vs. $r = 0.65$ for DunedinPoAm.

Consistent with the expectation that training the algorithm using more reliable probes would improve test-retest reliability, the reliable-probes-trained DunedinPACE showed superior test-retest reliability to the all-probes trained DunedinPACE in the test-retest reliability dataset published by Lehne et al. (GSE55763) (reliable-probes ICC = 0.96 vs 0.72 for all-probes). ICCs were similar for the two version of DunedinPoAm (reliable-probes ICC = 0.90 vs 0.89 for all probes).

