## [Editor Report]

This paper presents a new DNA methylation-based biomarker of aging: DunedinPACE. This biomarker is an updated version of DunedinPOAM, which was designed by the same group of authors to track an individual's Pace of Aging. It takes into account an additional measurement occasion (collected 20 years after inclusion) and only includes the most reliable DNA methylation probes, i.e. probes with little variation between technical replicates. DunedinPACE shows improved performance when compared to DunedinPOAM and can be used to complement previously generated DNA methylation-based biomarkers, such as GrimAge.

---

## [Decision Letter]

**Decision letter after peer review:**

Thank you for submitting your article "Quantification of the pace of biological aging in humans through a blood test: the DunedinPACE DNA methylation algorithm" for consideration by *eLife*. Your article has been reviewed by 3 peer reviewers, including Joris Deelen as the Reviewing Editor and Reviewer #3, and the evaluation has been overseen by Jessica Tyler as the Senior Editor. The following individual involved in review of your submission has agreed to reveal their identity: Matthew Suderman (Reviewer #2).

Essential revisions:

1. The reviewers all agree that it is unclear what the added value of DunedinPACE is over the previously generated DundinPOAM and, even more, the widely used GrimAge methylation biomarkers. This should be discussed more thoroughly and will require some additional analyses (see individual review reports).

2. The discussion between the reviewers also brought forward that the work by Levine and colleagues (doi: https://doi.org/10.1101/2021.04.16.440205), although shortly mentioned in the introduction, has not sufficiently been taken into account by the authors. This preprint has several implications for the current manuscript. First, the currently used methylation biomarkers, including GrimAge, will soon be updated using the correction proposed by Levine et al. and will outperform the versions used in the current manuscript. Thus, the iterative improvement in DunedinPACE vs DunedinPoAm may be the only relevant comparison. Second, the current manuscript uses test-retest of individual CpG probes to refine DunedinPACE. However Levine et al. show strong evidence that this strategy may improve performance for a single clock (e.g. better performance in DunedinPoAM to DunedinPACE) but that this may not generalize to other clocks. Rather, technical variability was best improved by PCA to extract a 'shared signal' vs individual CpGs. At minimum, the authors should consider various approaches to reduce technical reliability rather than focus solely on individual CpGs, including creating a methylation biomarker using principal components.

The authors should also address the additional points mentioned in the individual review reports below.

*Reviewer #1:*

This study details development of an improved iteration of a DNA methylation biomarker based on Pace of Aging measures developed in healthy, primarily Caucasian adults up to age 45 years in the highly unique longitudinal Dunedin Study. The new methylation biomarker is DunedinPACE; the predecessor DunedinPoAm was reported in Belsky et al. 2020 *eLife* 9,354870.

Strengths

The key contribution of this study is to address limitations in previous methylation biomarkers: (1) Pace of Aging measures now encompass a 20-year time span vs 12 years in Dunedin study; (2) training population includes adults up to 45 years (vs 38 years); and (3) methylation model is restricted only to probes that meet a minimum threshold for test re-test reliability.

1. DunedinPACE has good test-retest reliability, which is highly relevant to biomarker performance in clinical studies and interventions testing.

2. Not unexpectedly, the test-retest reliability is stronger for the new DunedinPACE vs. earlier DunedinPoAm, possibly owing to introduction of minimum probe reliability thresholds, though this is not explicitly tested.

3. The analyses presented generally follow established tests of criterion and construct validity across a set of cohort studies previously used to develop its predecessor DundedinPoAm (see Belsky et al. 2020 *eLife* 9,354870). The parallel strategy makes the seemingly complex set of validation studies a relatively straightforward comparison across biomarker iterations.

4. The effect sizes in cross-cohort validation analyses are larger for DunedinPACE relative to the foundational DunedinPoAm, and the effect sizes meet (GrimAge) or surpass effect sizes for other existing DNAm biomarkers. This supports strength of the new PACE biomarker iteration relative to most others.

5. Conclusions are generally well supported by data and authors refrain from extraneous discussion.

Weaknesses

1. Though several key limitations in DunedinPACE's predecessors were addressed, it is unfortunate that readers are not provided details regarding their relative contributions to overall biomarker performance.

2. It would be highly informative for future biomarker development to know if and to what extent population characteristics (cohort, age, or duration of observations for Pace of Aging measure) drive improved performance of DunedinPACE vs. DunedinPoAm relative to test-retest reliability or duration of longitudinal observation.

3. DunedinPACE performed similarly (reported effect sizes) to GrimAge, yet this equivalence is not addressed in discussion or conclusion. It is unclear under which conditions should one biomarker be prioritized over another if effect sizes are similar.

4. Forced language regarding DunedinPACE as a proxy for Pace of Aging muddles results presentation and section headings; this may inadvertently misrepresent select analyses.

5. Remaining concerns and suggestions are to improve readability, clarify nomenclature, and strengthen presentation of results, and do not influence primary conclusions or implications.

Title:

6. The title is regurgitated from the *eLife* 2020 publication but with a modification after the colon. This tedious for query building, search engines, and reference lists. The authors should thus consider adapting it.

Contribution of individual iterative changes:

7. To address individual contributions of each of the prior limitations, the authors could consider supporting analyses to address:

• Ddoes the inclusion of persons who are 45 years and now exhibiting detectable declines in health / function improve performance of the methylation biomarker, or is the time span (12 vs 20 years) the key determinant for calibration for improved performance?

– Potential approach: restrict model to population aged 32-45 years (~12 years, but older) and compare performance to current DunedinPACE and previous DunedinPoAm.

• Is minimum reliability threshold the key to improved performance or characteristics of the population used to calibrate the methylation biomarker?

– Potential approach: (a) revisit DunedinPoAm – restrict the existing biomarker to probes meeting reliability threshold, or (b) release test-retest reliability restrictions for probes included in DunedinPACE and evaluate.

Implications:

8. DunedinPACE is an iterative change to DunedinPoAm; clearly showing how each of these changes impact the resultant model bolsters significance.

9. DunedinPACE is tested head-to-head with previous derivations of DNAm clocks, but without insight regarding if or how changes leading to DunedinPACE could be introduced to rebuild better a GrimAge, PhenoAge, etc.

10. If primary determinant of improved performance is including aged >45 years, what does this mean for generalizability of clocks tuned in young/healthy population to trials in older adults or those in poorer health.

11. If analyses show that duration of longitudinal observation is key (20 years vs 12 years), then would future Dunedin derivations be expected to consistently outperform the previous versions and what would this mean for durability of the DunedinPACE as a biomarker?

GrimAge:

12. Addressing GrimAge vs DunedinPACE similarity in effect sizes may require some speculation, but would aid reader interpretation of results and could be simply handled with a short sentence or two in discussion.

Language – DunedinPACE as Pace of Aging Proxy:

13. Language referring to DunedinPACE and Pace of Aging is sometimes confusing, which could be attributed to the forced representation of DunedinPACE as a proxy for Pace of Aging.

• This results in instances of merged language (pg 12 "DunedinPACE Pace of Aging").

• The Results section and headings also suffer from forced association (marketing over reporting findings?). Examples:

– (Results pg 9) "DunedinPACE is indicates faster Pace of Aging in Chronologically Older Individuals." Results in text and Figure 3A show association with chronological age.

– (Results pg 10) "DunedinPACE shows faster Pace of Aging in individuals measured to be…" Forcing reference to Pace of Aging makes the heading cumbersome ("DunedinPACE is associated with other epigenetic clocks, measures of biologic age, and self-rated health").

– Authors are recommended to use simplicity and clarity throughout results and save comment about DunedinPACE as possible a proxy for Pace of Aging for the discussion.

Results, Text and Figure 3 Alignment:

– Page 9-10. "DunedinPACE indicates faster Pace of Aging in Chronologically Older.."

– The text opens with a sentences about mortality risk, but the only figure referenced is association with chronological age in Understanding Society study (Figure 3 Panel A). Recommend to omit or move to mortality risk references to Results section on NAS.

– The second paragraph provides information about exposure histories across birth cohorts. These results distract from the flow of results reporting in Figure 3 and subsequent section. While within-individual change may be supportive they are not essential and detract from readability.

For consideration (not mandatory): Naming Conventions

14. One improvement of DunedinPACE is the name; it is much easier to read / say compared with DunedinPoAm. However, would "PACE" or "PACEm" be sufficient? Adding "Dunedin…" seems to suggest that DunedinPACE or DunedinPoAm are specific to the foundational study of origin compared with GrimAge or PhenoAge.

15. "Faster" and "Slower" are used repeatedly to refer to higher vs. lower DunedinPACE (single-timepoint) compared to Pace of Aging (time component included). Using "faster" or "slower" may be appropriate for a Pace-proxy unlike most clocks. But referring to change in single timepoint DunedinPACE as "faster" is confusing, even if it is proposed as a proxy for Pace of Aging

*Reviewer #2:*

Although the paper is well-written, findings are not presented in a way that allows the reader to answer the most important question of the paper: how does DunedinPACE compare to DunedinPOAM? Little can be concluded from the improved performance in the Dunedin study data from which DunedinPACE and DunedinPOAM were both derived. In independent datasets, performance appears to be almost identical. Test-retest evaluations in the Sugden dataset similarly are difficult to interpret because probe reliability was calculated using the Sugden dataset.

In all evaluations using data used to derive DunedinPACE, authors should clearly state the potential for inflated performance. This is particularly true for comparisons with DundinPOAM which was trained on a subset of DunedinPACE data and previously published aging estimates Horvath/Hannum/Phenoage/Grimage trained entirely on external data.

*Reviewer #3:*

The manuscript by Belsky and colleagues reports the development of an improved methylation biomarker based on longitudinal health data (Pace of Aging) from the Dunedin Study. The authors have now used an additional wave of measurements and excluded methylation probes with low reliability based on recent publications. They subsequently associated their improved biomarker, which they called DunedinPACE, with many different health-related outcomes, such as physical functioning, morbidity and mortality. They show that DunedinPACE shows comparable results as GrimAge, a methylation biomarker based on mortality.

The major strength of the study is that the authors have managed to improve their previous methylation biomarker (DunedinPoAm) by adding additional data. They also show that DunedinPACE does reasonable well when associated with health-related outcomes. However, it is unclear what the added value of DunedinPACE is in comparison to GrimAge, because it does not outperform this previous biomarker in its association with any of the outcomes. Moreover, the provided data on the most relevant health-related outcomes, i.e. morbidity and mortality, is limited and it is unclear if the methylation biomarker would outperform other omics-based biomarkers (i.e. metabolomics and proteomics).

– The authors should highlight the added value of DunedinPACE in comparison to GrimAge. It looks like they do equally well in the analysis they performed, so what would be the advantage of using DunedinPACE over GrimAge, given that the latter is already widely used.

– The authors should also look at cause-specific mortality and morbidity to see if DunedinPACE reflects general health or is specific to one major disease, e.g. cardiovascular disease.

– The authors should test the predictive ability of DunedinPACE, ideally in comparison to currently used clinical biomarkers and biomarkers based on other omics data (see PMID: 34145379 and PMID: 31431621). If a direct comparison is not possible, the existence of these non-methylation biomarkers should still be mentioned, including their effect sizes for morbidity/mortality (if available).

– The authors should test the association of DunedinPACE with morbidity and mortality in different age-groups (i.e. stratify their data) to make sure the associations are not driven by early-life morbidity and mortality.

– It would be interesting if the authors can provide a biological interpretation (e.g. functional annotation / genetic association analysis) of the CpGs that are included in the DunedinPACE biomarker, as already done for some of the other (methylation-based) biomarkers (see for example PMID: 34187551 and PMID: 34038024).

---

## [Author Response]

Essential revisions:1. The reviewers all agree that it is unclear what the added value of DunedinPACE is over the previously generated DundinPOAM and, even more, the widely used GrimAge methylation biomarkers. This should be discussed more thoroughly and will require some additional analyses (see individual review reports).

Thank you for the opportunity to clarify value added of DunedinPACE. PoAm was the first iteration and PACE is an advance on it. Both were derived in the Dunedin Longitudinal Study. The value added of DunedinPACE over PoAm is primarily technical and consists of two key advances: (1) PACE offers improved precision of DNA methylation measurement of Pace of Aging; DunedinPACE is nearly twice as accurate as DunedinPoAm in predicting longitudinal Pace of Aging (correlations of r=0.6 and r=0.8 translate into coefficients of determination of R^2^=0.36 and 0.64). (2) PACE offers improved technical reliability of the DNA methylation measurement (ICC>.95 in technical replicates). High reliability is essential for detection of change over time in RCTs. These two advances are reviewed in the first and second paragraphs of the discussion.

The value added of DunedinPACE over DNAmethylation clocks is in its design. Many features of a person’s biology may predict their risk of disease and death. The 2^nd^-generation PhenoAge and GrimAge clocks operationalize these. But only some of these features are aging. DunedinPACE operationalizes aging itself as decline in physiological integrity over years. DunedinPACE has three design features that isolate aging signal:

(1) The Dunedin cohort were all born in the same year. This eliminates bias from historical-exposure effects, including differences between people born in different years in their mothers’ health and environment when they were in-utero, their childhood nutrition, their exposure to childhood infections, and their exposure to environmental toxicants, such as air pollution from leaded gasoline and second-hand tobacco smoke in the workplace.

(2) Dunedin cohort members were young adults at the time of baseline measurements. This eliminates survival bias; none of the cohort had died of aging-related causes by the time of baseline Pace of Aging assessment. It also separates aging from age-related disease; physiological changes observed beginning in cohort members’ 20s represent pre-clinical changes in system integrity that are causes of disease, not disease sequelae.

(3) DunedinPACE is modeled on decline in the integrity of multiple organ systems during adult life. This isolates aging-related changes from deficits established early in development.

In contrast, clocks were developed on mixed-age populations and cannot distinguish aging from historical-exposure effects. Clocks were developed from mixed-age or older populations, introducing survival bias and uncertainty about what is aging versus disease. Finally, clocks are based on a single cross-sectional measurement and do not distinguish aging-related changes from deficits established early in development. We have added a final paragraph to the introduction to clarify these points.

“As a new DNA methylation measure of biological aging, DunedinPACE joins a well-established battery of DNA methylation measures known as “clocks”, so-named for their accurate prediction of chronological age (Horvath and Raj, 2018). […] DunedinPACE, therefore, represents a complementary tool for DNA methylation quantification of biological aging.”

Consistent with these distinguishing features, DunedinPACE captures information about aging that is not measured by the clocks. We have added a series of multivariate analyses in the Framingham Heart Study cohort (which was used to develop GrimAge) that show DunedinPACE predicts mortality, incident cardiovascular disease, and incident disability over and above the clocks, including GrimAge. This is a rigorous test since GrimAge was developed using the Framingham data. Results are reported in the final paragraph of the Results section and detailed in Supplementary file 1C.

“Incremental predictive value of DunedinPACE in an analysis of morbidity, disability, and mortality. DunedinPACE is distinct from DNA methylation clocks in the method of its development and in its interpretation. […] Results for all models are reported in Supplementary file 1C.”

2. The discussion between the reviewers also brought forward that the work by Levine and colleagues (doi: https://doi.org/10.1101/2021.04.16.440205), although shortly mentioned in the introduction, has not sufficiently been taken into account by the authors. This preprint has several implications for the current manuscript. First, the currently used methylation biomarkers, including GrimAge, will soon be updated using the correction proposed by Levine et al. and will outperform the versions used in the current manuscript. Thus, the iterative improvement in DunedinPACE vs DunedinPoAm may be the only relevant comparison. Second, the current manuscript uses test-retest of individual CpG probes to refine DunedinPACE. However Levine et al. show strong evidence that this strategy may improve performance for a single clock (e.g. better performance in DunedinPoAM to DunedinPACE) but that this may not generalize to other clocks. Rather, technical variability was best improved by PCA to extract a 'shared signal' vs individual CpGs. At minimum, the authors should consider various approaches to reduce technical reliability rather than focus solely on individual CpGs, including creating a methylation biomarker using principal components.

Thank you for this suggestion. First, as our paper reports, DunedinPACE already demonstrates excellent test-retest reliability (ICC=0.96-0.97). So, the potential for the approach proposed in the Higgins-Chen et al. preprint to improve the test-retest reliability of our measure is virtually nil. Moreover, the method introduced by Higgins-Chen was designed to address the limitation that, when DNAm clocks were re-trained using subsets of more reliable probes, the resulting clock measures were less predictive of morbidity and mortality as compared to their original versions. As our data show, that is not the case for the reliable-probes-based DunedinPACE; DunedinPACE effect-sizes are the same as or larger than effect-sizes for the original DunedinPoAm, which did not use reliable probes. Thus, the Higgins-Chen et al. approach may be a useful a solution to a thorny problem that uniquely afflicts clocks such as PhenoAge, but that problem does not afflict DunedinPACE and the corrective approach is not needed by DunedinPACE.

However, we agree that it is important to acknowledge that the analysis by Higgins-Chen et al. suggests that the approach we used does not generalize to existing epigenetic clocks. Therefore, in addition to the text in our article which already acknowledges the potential for methods such as those proposed in the Higgins-Chen et al. preprint to improve reliabilities of existing clocks, we now also acknowledge that the method we used may not generalize to the epigenetic clocks.

“Comparison of DunedinPACE reliability to reliabilities for the original DunedinPoAm and the DNA methylation clocks are shown in Supplemental Figure 2. DunedinPACE is as reliable or more so as compared to the GrimAge clock and more reliable in comparison to the other clocks, although methods have been proposed that may improve reliabilities for the other clocks (Higgins-Chen et al., 2021).” (results p10 para 4)

“Interestingly, in contrast to reports for some epigenetic clocks (Higgins-Chen et al., 2021), the restriction of machine learning analysis to a more-reliably-measured subset of CpG sites on the Illumina EPIC array did not harm criterion validity of DunedinPACE; effect-sizes for prediction of morbidity and mortality were as large or larger in comparison to DunedinPoAm.” (discussion p17 para 2)

The authors should also address the additional points mentioned in the individual review reports below.Reviewer #1:This study details development of an improved iteration of a DNA methylation biomarker based on Pace of Aging measures developed in healthy, primarily Caucasian adults up to age 45 years in the highly unique longitudinal Dunedin Study. The new methylation biomarker is DunedinPACE; the predecessor DunedinPoAm was reported in Belsky et al. 2020 eLife 9,354870.StrengthsThe key contribution of this study is to address limitations in previous methylation biomarkers: (1) Pace of Aging measures now encompass a 20-year time span vs 12 years in Dunedin study; (2) training population includes adults up to 45 years (vs 38 years); and (3) methylation model is restricted only to probes that meet a minimum threshold for test re-test reliability.1. DunedinPACE has good test-retest reliability, which is highly relevant to biomarker performance in clinical studies and interventions testing.2. Not unexpectedly, the test-retest reliability is stronger for the new DunedinPACE vs. earlier DunedinPoAm, possibly owing to introduction of minimum probe reliability thresholds, though this is not explicitly tested.3. The analyses presented generally follow established tests of criterion and construct validity across a set of cohort studies previously used to develop its predecessor DundedinPoAm (see Belsky et al. 2020 eLife 9,354870). The parallel strategy makes the seemingly complex set of validation studies a relatively straightforward comparison across biomarker iterations.4. The effect sizes in cross-cohort validation analyses are larger for DunedinPACE relative to the foundational DunedinPoAm, and the effect sizes meet (GrimAge) or surpass effect sizes for other existing DNAm biomarkers. This supports strength of the new PACE biomarker iteration relative to most others.5. Conclusions are generally well supported by data and authors refrain from extraneous discussion.Weaknesses1. Though several key limitations in DunedinPACE's predecessors were addressed, it is unfortunate that readers are not provided details regarding their relative contributions to overall biomarker performance.2. It would be highly informative for future biomarker development to know if and to what extent population characteristics (cohort, age, or duration of observations for Pace of Aging measure) drive improved performance of DunedinPACE vs. DunedinPoAm relative to test-retest reliability or duration of longitudinal observation.

Thank you for these suggestions. We have conducted additional analyses to refine inference about which aspects of DunedinPACE development contributed to the key improvements over DunedinPoAm, i.e. improved fit to the Pace of Aging criterion and improved test-retest reliability. Specifically, we created two new algorithms: (1) a new version of DunedinPACE trained to fit 20-year Pace of Aging from all probes shared on the EPIC and 450k arrays (referred to as “all probes DunedinPACE”); and (2) a version of DunedinPoAm trained from the subset of probes meeting the reliability threshold set by Sugden. (referred to as “reliable probes DunedinPoAm”). Analysis of criterion fit to Pace of Aging and test-retest reliability are reported in Supplemental Results and mentioned in the Results section (p11, para 1). There are two key findings from this analysis: First, all versions trained on 20-year, 4-time-point Pace of Aging show higher correlation with their Pace of Aging criterion as compared to all versions trained on 12-year, 3-time-point Pace of Aging. This result indicates that the improved criterion fit of DuneindPACE comes from inclusion of additional follow-up in the Pace of Aging measure. Second, all algorithms trained on the subset of reliable probes showed higher test-retest reliability as compared with all algorithms trained in the complete set of probes. This result indicates that the improved test-retest reliability of DunedinPACE comes from the restriction of elastic net regression analysis to the subset of reliable probes.

3. DunedinPACE performed similarly (reported effect sizes) to GrimAge, yet this equivalence is not addressed in discussion or conclusion. It is unclear under which conditions should one biomarker be prioritized over another if effect sizes are similar.

Thank you for the opportunity to clarify. We view DunedinPACE as a measure complementary to GrimAge. As explained in our response to Essential Revisions 1, DunedinPACE is distinct from GrimAge in its design and its meaning. In addition, it also captures unique information about risk for disease and death; i.e. DunedinPACE improves prediction of morbidity and mortality over and above GrimAge. This is detailed in the final paragraph of the Results section and in Supplementary file 1C.

4. Forced language regarding DunedinPACE as a proxy for Pace of Aging muddles results presentation and section headings; this may inadvertently misrepresent select analyses.

Done. Thank you for bringing this to our attention. The text has been revised for clarity.

5. Remaining concerns and suggestions are to improve readability, clarify nomenclature, and strengthen presentation of results, and do not influence primary conclusions or implications.Title:6. The title is regurgitated from the eLife 2020 publication but with a modification after the colon. This tedious for query building, search engines, and reference lists. The authors should thus consider adapting it.

Done. The new title is “DunedinPACE: A DNA methylation biomarker of the Pace of Aging”.

Contribution of individual iterative changes:7. To address individual contributions of each of the prior limitations, the authors could consider supporting analyses to address:• Does the inclusion of persons who are 45 years and now exhibiting detectable declines in health / function improve performance of the methylation biomarker, or is the time span (12 vs 20 years) the key determinant for calibration for improved performance?– Potential approach: restrict model to population aged 32-45 years (~12 years, but older) and compare performance to current DunedinPACE and previous DunedinPoAm.• Is minimum reliability threshold the key to improved performance or characteristics of the population used to calibrate the methylation biomarker?– Potential approach: (a) revisit DunedinPoAm – restrict the existing biomarker to probes meeting reliability threshold, or (b) release test-retest reliability restrictions for probes included in DunedinPACE and evaluate.Implications:8. DunedinPACE is an iterative change to DunedinPoAm; clearly showing how each of these changes impact the resultant model bolsters significance.9. DunedinPACE is tested head-to-head with previous derivations of DNAm clocks, but without insight regarding if or how changes leading to DunedinPACE could be introduced to rebuild better a GrimAge, PhenoAge, etc.

This is an interesting idea. But, lacking the data used to train the other clocks, we are unable to conduct the same analysis. We now note that the Higgins-Chen preprint explored restriction of CpG sites based on ICC criteria for good test-retest reliability and found this approach was generally harmful; it reduced the criterion validity of their clocks. However restriction of CpG sites to reliable sites did not reduce criterion validity for DunedinPACE (Discussion p17 para 2, text quoted above in response to Essential Revision 2).

10. If primary determinant of improved performance is including aged >45 years, what does this mean for generalizability of clocks tuned in young/healthy population to trials in older adults or those in poorer health.

Thank you for the opportunity to clarify. Prediction of morbidity and mortality in older adults in the Normative Aging Study and Framingham cohorts in our paper is similar for DunedinPoAm (developed from Pace of Aging measured over ages 26-38) as for DunedinPACE (developed from Pace of Aging measured over ages 26-45). Therefore, the age range used in measuring Pace of Aging does not seem to be a critical feature of prediction of morbidity and mortality in older adults. DunedinPACE, though trained on data from age 26 to 45, predicts morbidity and mortality in cohorts in their 70s and 80s.

11. If analyses show that duration of longitudinal observation is key (20 years vs 12 years), then would future Dunedin derivations be expected to consistently outperform the previous versions and what would this mean for durability of the DunedinPACE as a biomarker?

We expect that, when another wave of data from the Dunedin Study is collected in 5-7 years, a new Pace of Aging measurement will be evaluated. At that time, we will be able to clarify whether longer follow-up intervals extending into later stages of life contribute new and important information to a Pace of Aging DNA methylation measure. Until then, geroscience needs the best measurement tools available to evaluate intervention effects today. DunedinPACE is an improvement over what is already available.

GrimAge:12. Addressing GrimAge vs DunedinPACE similarity in effect sizes may require some speculation, but would aid reader interpretation of results and could be simply handled with a short sentence or two in discussion.

Thank you for this suggestion. We have added text to the introduction (p7-8), results (p15-16), and discussion (p17-18). These changes are also summarized in our response to Essential Revision 1 above.

Language – DunedinPACE as Pace of Aging Proxy:13. Language referring to DunedinPACE and Pace of Aging is sometimes confusing, which could be attributed to the forced representation of DunedinPACE as a proxy for Pace of Aging.• This results in instances of merged language (pg 12 "DunedinPACE Pace of Aging").• The Results section and headings also suffer from forced association (marketing over reporting findings?). Examples:– (Results pg 9) "DunedinPACE is indicates faster Pace of Aging in Chronologically Older Individuals." Results in text and Figure 3A show association with chronological age.– (Results pg 10) "DunedinPACE shows faster Pace of Aging in individuals measured to be…" Forcing reference to Pace of Aging makes the heading cumbersome ("DunedinPACE is associated with other epigenetic clocks, measures of biologic age, and self-rated health").– Authors are recommended to use simplicity and clarity throughout results and save comment about DunedinPACE as possible a proxy for Pace of Aging for the discussion.

Done. We have revised text and figure titles.

Results, Text and Figure 3 Alignment:– Page 9-10. "DunedinPACE indicates faster Pace of Aging in Chronologically Older.."– The text opens with a sentences about mortality risk, but the only figure referenced is association with chronological age in Understanding Society study (Figure 3 Panel A). Recommend to omit or move to mortality risk references to Results section on NAS.

Thank you for the opportunity to clarify. The text referencing mortality risk explains why we anticipate a correlation between DunedinPACE and chronological age. We have further revised the text for clarity.

“In demography, the pace of biological aging in a population can be estimated from the rate of increase in mortality risk from younger to older chronological ages (Hägg et al., 2019). In humans and many other species, the increase in mortality risk with advancing chronological age is nonlinear, suggesting that the pace of biological aging accelerates as we grow older (Gompertz, 1825; Kirkwood, 2015; Olshansky and Carnes, 1997). We tested if, consistent with this hypothesis, DunedinPACE was faster in chronologically older as compared to younger individuals in data from the Understanding Society Study (n=1175, age range 28-95). Chronologically older Understanding Society participants had faster DunedinPACE as compared to younger ones (r=0.32, Figure 3 Panel A). This correlation was threefold larger as compared to the correlation for the original DunedinPoAm (r=0.11). ” (p11 para 3)

– The second paragraph provides information about exposure histories across birth cohorts. These results distract from the flow of results reporting in Figure 3 and subsequent section. While within-individual change may be supportive they are not essential and detract from readability.

Thank you for this suggestion. We think these data on within-individual change are essential for establishing the independence of DunedinPACE correlations with chronological age from historical exposures that vary by birth year and may alter the methylome (i.e. “cohort effects”). While cohort effects are not yet widely appreciated as a source of bias in biology, they are well established in human epidemiologic research. Therefore, we have retained the text.

For consideration (not mandatory): Naming Conventions14. One improvement of DunedinPACE is the name; it is much easier to read / say compared with DunedinPoAm. However, would "PACE" or "PACEm" be sufficient? Adding "Dunedin…" seems to suggest that DunedinPACE or DunedinPoAm are specific to the foundational study of origin compared with GrimAge or PhenoAge.

Thank you for this suggestion. We appreciate the inclusion of the name of the study in the name of our measure is somewhat cumbersome. But we also think it is essential. Data sources used in the derivation of machine-learning-based measures can affect the measure in many ways – something we acknowledge in the limitations section of our discussion. In our view, we should all think of the Framingham-GrimAge, the InCHIANTI-PhenoAge, etc. We have therefore elected to retain our original study name to inform the science. In addition, there is an ethical reason. Dunedin Study participants were consulted in focus groups about their views on using Study data to derive a measure that might have intellectual property, and they asked that their contribution of repeated blood samples be acknowledged and recognized in the naming of the measure. Given the current unpleasant situation with the family of Henrietta Lacks in reaction to commercial use of HELA cells, consulting the Study participants was good policy.

15. "Faster" and "Slower" are used repeatedly to refer to higher vs. lower DunedinPACE (single-timepoint) compared to Pace of Aging (time component included). Using "faster" or "slower" may be appropriate for a Pace-proxy unlike most clocks. But referring to change in single timepoint DunedinPACE as "faster" is confusing, even if it is proposed as a proxy for Pace of Aging.

Thank you for this suggestion. We have retained the language of faster/slower as DunedinPACE is not a clock and does not measure an “age” value. DunedinPACE is a prediction of Pace of Aging. Higher values correspond to faster Pace of Aging and lower values to slower Pace of Aging. Faster/slower therefore aligns with the measure. In addition, this precise language helps to distinguish DunedinPACE from results for the clocks, which were designed to generate values younger/older, but not faster/slower.

Reviewer #2:Although the paper is well-written, findings are not presented in a way that allows the reader to answer the most important question of the paper: how does DunedinPACE compare to DunedinPOAM? Little can be concluded from the improved performance in the Dunedin study data from which DunedinPACE and DunedinPOAM were both derived. In independent datasets, performance appears to be almost identical. Test-retest evaluations in the Sugden dataset similarly are difficult to interpret because probe reliability was calculated using the Sugden dataset.

We report head-to-head comparisons of DunedinPACE with DunedinPoAm in the text and in the supplemental figures and specifically note the outcomes of these comparisons in the Results section. We have highlighted these comparisons in the revised submission.

Regarding Test-retest reliability: the probe reliability threshold used to define the CpG set used to develop DunedinPACE was defined in only one of the datasets used to evaluate test-retest reliability (Sugden B n=350 450k-EPIC, Supplemental Figure 2). Results are consistent across all datasets, arguing against the finding being an artifact of the Sugden dataset.

In all evaluations using data used to derive DunedinPACE, authors should clearly state the potential for inflated performance. This is particularly true for comparisons with DundinPOAM which was trained on a subset of DunedinPACE data and previously published aging estimates Horvath/Hannum/Phenoage/Grimage trained entirely on external data.

Done. This issue applies to the Dunedin data, but not to the several other datasets analyzed. We now note that the analysis of physical and cognitive functioning and subjective signs of aging within the Dunedin cohort may result in some upward bias to effect-size estimates for DunedinPACE and DunedinPoAm within that cohort.

“DunedinPACE and DunedinPoAm were developed from analysis of Pace of Aging in the Dunedin cohort; effect-sizes for these measures may be over-estimated within this cohort.” (p10 para 2)

Reviewer #3:The manuscript by Belsky and colleagues reports the development of an improved methylation biomarker based on longitudinal health data (Pace of Aging) from the Dunedin Study. The authors have now used an additional wave of measurements and excluded methylation probes with low reliability based on recent publications. They subsequently associated their improved biomarker, which they called DunedinPACE, with many different health-related outcomes, such as physical functioning, morbidity and mortality. They show that DunedinPACE shows comparable results as GrimAge, a methylation biomarker based on mortality.The major strength of the study is that the authors have managed to improve their previous methylation biomarker (DunedinPoAm) by adding additional data. They also show that DunedinPACE does reasonable well when associated with health-related outcomes. However, it is unclear what the added value of DunedinPACE is in comparison to GrimAge, because it does not outperform this previous biomarker in its association with any of the outcomes. Moreover, the provided data on the most relevant health-related outcomes, i.e. morbidity and mortality, is limited and it is unclear if the methylation biomarker would outperform other omics-based biomarkers (i.e. metabolomics and proteomics).– The authors should highlight the added value of DunedinPACE in comparison to GrimAge. It looks like they do equally well in the analysis they performed, so what would be the advantage of using DunedinPACE over GrimAge, given that the latter is already widely used.

Thank you for this suggestion. As explained in response to Essential Revision 1 and in a new paragraph added to the introduction (p7-8), DunedinPACE represents distinct information about aging from GrimAge. Moreover, empirically, DuedinPACE is an independent predictor of morbidity, disability, and mortality (Results p15-16; Supplemental Table 5). Therefore, DunedinPACE is complementary to GrimAge (Discussion p18 para 2).

– The authors should also look at cause-specific mortality and morbidity to see if DunedinPACE reflects general health or is specific to one major disease, e.g. cardiovascular disease.

Thank you for this suggestion. We have added time-to-event analysis of cardiovascular disease and stroke and repeated-measures analysis of incident disability in the Framingham Heart Study Cohort. See results p14 para 1, Supplemental Table 4, and Supplemental Figure 4 panels C and D.

We did not include analysis of cause-specific mortality because the number of deaths from individual causes is too small in the datasets available to us for confident ascertainment of differential associations with different causes of death. We now acknowledge this as a limitation (Discussion p18 para 2).

In response to the reviewer’s request, we did conduct the analysis using the data available from the Framingham Heart Study (which classifies deaths as due to Coronary Heart Disease, other Cardiovascular Disease, and all other causes). The effect-size for cardiovascular disease is HR=1.46 95% CI [1.23-1.72]. The effect-size for non-CVD causes is HR=1.70 95% CI [1.55-1.87].

– The authors should test the predictive ability of DunedinPACE, ideally in comparison to currently used clinical biomarkers and biomarkers based on other omics data (see PMID: 34145379 and PMID: 31431621). If a direct comparison is not possible, the existence of these non-methylation biomarkers should still be mentioned, including their effect sizes for morbidity/mortality (if available).

Thank you for this suggestion. We now acknowledge the exciting progress in proteomics and metabolomics research on aging in the limitations section of the discussion:

“Recent reports suggest valuable information about aging and healthspan may also be captured in metabolomic and proteomic datasets (Deelen et al., 2019; Eiriksdottir et al., 2021; Jansen et al., 2021; Lehallier et al., 2019; Tanaka et al., 2020), although such datasets are as yet rare. Future studies should compare DunedinPACE to measures derived from these biological levels of analysis.” (p18 para 2)

– The authors should test the association of DunedinPACE with morbidity and mortality in different age-groups (i.e. stratify their data) to make sure the associations are not driven by early-life morbidity and mortality.

Thank you for the opportunity to clarify. All mortality analyses were conducted using data from older adults (Normative Aging Study men were aged 77 years on average at DNA methylation baseline; Framingham participants were aged 66 years on average a DNA methylation baseline; see results p13-14). Therefore, early-life morbidity and mortality cannot explain the results observed.

– It would be interesting if the authors can provide a biological interpretation (e.g. functional annotation / genetic association analysis) of the CpGs that are included in the DunedinPACE biomarker, as already done for some of the other (methylation-based) biomarkers (see for example PMID: 34187551 and PMID: 34038024).

We have conducted the analysis suggested by the reviewer and results are included below. However, it is our preference not to include these in the manuscript. The reason is that, in our view, this type of bioinformatic annotation is inappropriate to interpretation of CpG sets obtained from machine-learning analysis. The elastic net algorithm we used to derive DunedinPACE is ignorant of biology. It simply chooses the minimum set of sites that, together, maximize prediction of the Pace of Aging criterion. For example, sites that are highly correlated statistically but which are located near different genes and therefore have potentially distinct biological interpretations are, from the view of the elastic net, interchangeable.

We note that the articles cited performed analysis on CpG sets compiled from many different measures of aging. In our view, this is a substantively different type of analysis than what we can conduct with the single set of probes included in DunedinPACE.

Pathway Analysis of DunedinPACE probes October 2021

We performed pathway analysis using the Ingenuity Pathway Analysis tool (IPA, Qiagen). We did this in three steps: First, we assigned DunedinPACE DNA methylation probes to the most proximal gene (in base pairs) based on genome assembly hg19. Second, we uploaded this list of genes to IPA, which mapped to 140 HUGO gene names (33 were unmapped and represented non-coding genes/ESTs). Of these, 7 gene names were duplicated (i.e. two DNA methylation probes mapped to the same gene); after removing the duplicates, 133 gene names were left for analysis. Third, we performed pathway analysis, restricting the search to (a) pathways known in humans, and (b) pathways experimentally observed in blood cells.

This analysis identified 9 canonical pathways which were enriched for genes in our list (nominal *p-value* cut-off = 0.05), with enrichment scores ranging from 1.29-1.86. These pathways were described by 12 of the 133 genes; Four genes were enriched in multiple pathways (*GAD1*, 4 pathways; *ADCY8*, 4 pathways; *KCNQ2*,2 pathways; *SOCS3*, 2 pathways).

Initial pathway analyses indicate that GABA synthesis and cell signaling might be pathways of interest with respect to DunedinPACE. However, there are some caveats to this conclusion. First, we selected the gene associated with a DNA methylation probe based on physical proximity; transcriptional relationships between genes and DNA methylation probes could be driven by factors other than physical distance between them. Such functional annotation could lead to gene assignation that differs from the one used here. Second, we do not have information on the relationship between the level of probe methylation and gene transcription; inclusion of this information could impact the assigned probabilities associated with pathway enrichment. The third caveat is that DunedinPACE is a statistical construct, not a biological one and, as such, inferences made about individual probes and their biological function may not be appropriate. It is important to remember that DunedinPACE, like PhenoAge and GrimAge, is an efficient algorithm that allows other researchers, who do not have the trained-upon data, to measure the phenotype of interest (in our case, the Pace of Aging which is made up of 20 years of repeatedly measured multiple biomarkers; in the case of GrimAge, a suite of plasma proteins, etc.). These algorithms comprise statistically efficient probes that capture a phenotype that would otherwise be inaccessible. Because their criteria for selection into the algorithm are based in statistics and not function, it is not appropriate to think of the resulting probes in biological terms. Moreover, DunedinPACE does not select every probe that is associated with the Pace of Aging. Rather, the elastic net has selected out probes that best predict DunedinPACE while minimizing both variance and bias in estimation. A biologically meaningful analysis could be performed through an Epigenome-Wide Association Study (EWAS) of the Pace of Aging. (The set of CpG sites with significant univariate associations with Pace of Aging is not equivalent to the set of elements identified by elastic net regression as most efficient for predicting Pace of Aging.) For these reasons, we are happy to share the results reported above in this response, but would prefer to not include them in the manuscript.